# Monitoring of compliance with fuel sulfur content regulations through UAV measurements of ship emissions

Fan Zhou[1], Shengda Pan[1], Wei Chen[2], Xunpeng Ni[2], Bowen An[1]

[1.]College of Information Engineering, Shanghai Maritime University
[2.]Pudong Maritime Safety Administration of the People's Republic of China

*Correspondence to*: Fan Zhou (fanzhou_cv@163.com)

**Abstract.** Air pollution from ship exhaust gas can be reduced by the establishment of Emission Control Areas (ECAs). Efficient supervision of ship emissions is currently a major concern of maritime authorities. In this study, an Unmanned Aerial Vehicle (UAV)-based measurement system for exhaust gas from ships was designed and developed. Sensors were mounted on the UAV to measure the concentrations of $SO_2$ and $CO_2$ in order to calculate the fuel sulfur content (FSC) of ships. The Waigaoqiao port in the Yangtze River Delta, an ECA in China, was selected for monitoring compliance with FSC regulations. Unlike in situ or airborne measurements, the proposed measurement system could be used to determine the smoke plume at about 5 m from the funnel mouth of ships, thus providing a means for estimating the FSC of ships. In order to verify the accuracy of these measurements, fuel samples were collected at the same time and sent to the laboratory for chemical examination, and these two types of measurements were compared. After 23 comparative experiments, the results showed that, in general, the deviation of the estimated value for FSC was less than 0.03% (m/m) at an FSC level ranging from 0.035% (m/m) to 0.24% (m/m). Hence, UAV measurements can be used for monitoring of ECAs for compliance with FSC regulations.

## 1. Introduction

With the rapid development of international shipping in recent years, air pollution caused by ship emissions has become serious. Estimations show that ships contribute 4-9% of global $SO_2$ emissions and 15% of $NO_x$ (Eyring et al., 2010). According to the United Nations Conference on Trade And Development (UNCTAD, 2017), the volume of the world's seaborne trade grew by 66% between 2000 and 2015. As global commerce expands, ocean-going ships consume more fuel, generally low-quality residual fuel containing high concentrations of sulfur and heavy metals (Lack et al., 2011). From the viewpoint of spatial distribution, the highest emissions of $SO_2$ per unit area occur in the eastern and southern China seas, sea areas in south-eastern and southern Asia, Red Sea, Mediterranean Sea, North Atlantic near the European coast, Gulf of Mexico and Caribbean Sea, and along the western coast of North America (Johansson et al., 2017). Liu et al. (2016) reported that East Asia accounted for 16% of global shipping $CO_2$ emissions in 2013, which was an increase compared to only 4–7% in 2002–2005. In the research of Russo et al. (2018), who evaluated the contribution of shipping to overall emissions over

Europe, this sector was found to represent on average 16%, 11%, and 5% of the total $NO_x$, $SO_x$, and $PM_{10}$ emissions, respectively.

In order to limit hazards caused by ship emissions, the International Maritime Organization (IMO) extended the MARPOL 73/78 International Convention for the Preventions for Pollution of Air Pollution from Ship (MARPOL, 1997). In 2005,
some regulations went into effect after being accepted by appropriate laws of the signatory states (at the European level it was received with the directives 1999/32/EC, 1999, and 2005/33/EC, 2005), and introduces limits to marine fuel sulfur content and engine performance to reduce $SO_x$ and $NO_x$ emissions. Further amendments to Annex VI were adopted in 2008 and entered into force in 2010. Fuel sulfur content (FSC) is normally given in units of percent sulfur content by mass; in the following written as % (m/m). Following the IMO regulation, the global cap for FSC in marine fuel was set in 2012 at 3.5%
(m/m), and it will be reduced to 0.5% (m/m) by 2020. In addition, the IMO provides for the establishment of Emission Control Areas (ECAs) to control ship emissions, where there are more stringent controls on ship emissions. At present, the Baltic Sea, the North Sea, the North American area, and the United States Caribbean Sea are designated as ECAs (IMO, 2017). The FSC limit was set to 0.1% (m/m) in those areas beginning in 2015.

China is one of the world's busiest and fastest-growing shipping regions. In 2016, China accounted for seven of the world's
top 10 ports and 11 of the top 20. In order to reduce the air pollution caused by ship emissions, the Atmospheric Pollution Prevention and Control Law of the People's Republic of China was promulgated in 2015 (Standing Committee of the National People's Congress, 2015). Three domestic emission control areas (DECA) were set up, which include the Yangtze River Delta, the Pearl River Delta, and Bohai Rim (Beijing-Tianjin-Hebei Region). The current stage of the plan requires that the FSC does not exceed 0.5% (m/m).
With the above regulations in place, the main question remains on how to efficiently verify compliance of ships in the ECAs with the regulation. At present, the most accurate method for checking compliance is to collect fuel samples from ships at berth by state port control authorities, and then analyze the samples at certified laboratories or by portable detectors. However, it is time consuming and few ships are effectively controlled. Another problem is that sailing ships within the ECAs are not checked.
Several studies have suggested inferring FSC by monitoring ship emissions, and then identifying ships with excessive FSC. According to the available literature, these approaches include optical methods (LIDAR (Fan et al., 2018), Differential Optical Absorption Spectroscopy (DOAS) (Seyler et al., 2017), UV camera (Prata, 2014)) or "sniffer" methods (Balzani Lööv et al., 2014, Beecken et al., 2015). Optical methods analyze the variation of the light properties after interaction with the exhaust plume and allow, if the local wind field is known, to determine the emission rate of $SO_2$. The
simultaneous measurement of $CO_2$ and $SO_2$ emissions at a routine basis with these systems is unrealistic at the moment (Balzani Lööv et al., 2014). Thus, the amount of fuel burned at the time of measurement is unknown and has to be estimated via modeling to calculate the FSC. For instance, the model STEAM (ship traffic emission assessment model), developed by the Finnish Meteorological Institute (Jalkanen et al., 2009) was used in the research for estimating FSC by Balzani Lööv et al. (2014). In addition, using the ratio of $SO_2$ and $NO_2$ measured via DOAS in the ship' plume can be used as

an indicator of FSC (Johan, R et al. 2017, Cheng, Y et al, 2019). The advantage of the optical method is that it can detect ship emissions at a long distance (thousands of meters away), but it is limited in that it can only distinguish between a high FSC (>1% (m/m)) and a low FSC (<1% (m/m)) (Johan et al., 2017). The "sniffing" methods are based on simultaneous measurement of elevated $SO_2$ and $CO_2$ concentrations in the exhaust plume from the target ship and comparing them with the

background. The measurement of $CO_2$ allows for relating the measurement of $SO_2$ to the amount of fuel burned at a given time, thus enabling the calculation of FSC directly. The concentration of $SO_2$ in plumes was generally measured using UV fluorescence sensors, and $CO_2$ was measured using a non-dispersive infrared analyzer (NDIR) or cavity ring down spectrometer (CRDS). The advantage of the "sniffing" method is that it offers more accurate estimation for FSC. However, the instrument must be placed in the plume exhausted by the target ship. In some studies (Van Roy and Scheldeman, 2016a,

2016b), the "sniffing" method offers a measurement accuracy between 0.1–0.2% (m/m) FSC, which can be further increased up to 0.05–0.1% (m/m) FSC if combined with an additional $NO_x$ sensor. This is because the response of $SO_2$ analyzers (fluorescence) has cross sensitivity to NO. Deviations are not the same at different FSC levels, with an estimated relative uncertainty of 20% (m/m) for ships with 1% (m/m) FSC and a relative uncertainty of 50–100% at 0.1% (m/m) FSC. Balzani Lööv et al. (2014) obtained the following FSC measurements based on the "sniffer" principle: 0.86±0.23% (m/m) from land,

1.2±0.15% (m/m) from an on-board stack, and 1.13±0.18% (m/m) from a mobile platform. There was a 6% relative uncertainty for an FSC of 1% (m/m) but a 60% relative uncertainty for an FSC of 0.1% (m/m). It is important to note that the accuracy of the results of monitoring is a difficult issue to address, and the accuracy of estimates in the literature may not always be comparable. For ideal comparison results, one would need to board the ship to take fuel samples, which is particularly difficult for sailing ships.

Ship emission measurements can be divided into land-based (Kattner et al., 2015, Yang et al., 2016), airborne-based (Beecken et al., 2014, Aliabadi et al., 2016), marine-based (Cappa et al., 2014), satellite-based (Ding et al., 2018) and Unmanned Aerial Vehicle (UAV)-based (Villa et al., 2019) according to different platforms. Land-based measurements provide continuous observation but are greatly affected by wind speed, wind direction, and the distance between the ship and equipment. Airborne-based measurements can approach ship's plume and collect exhaust gas from the target ship. However,

the cost of airborne platforms is high, and it requires active sampling of ship exhaust plumes at low altitude. The closer the detector is to the ship's plume, the more accurate the results. However, safety risks are also relatively high near the plume. Marine-based measurements are suitable for studying the discharge from individual ships. The monitoring equipment is generally installed and used by research institutions or ship owners. This is not subjected to FSC inspection by government regulatory authorities. Satellite-based measurements are suitable for large-scale observation and mainly used to observe the

$NO_x$ emissions of ships. UAV-based measurements have gradually increased in the research regarding the atmosphere (Malaver Rojas et al., 2015, Mori et al., 2016). However, to date, there are relatively few applications of these measurements in ship emissions. As such, the most suitable approach for monitoring compliance is to employ "sniffer" measurements taken by aircraft. Optical measurements and "sniffer" measurements of gases in the exhaust plume of ships and more details on

such measurements can be found in several related papers (Balzani Lööv et al., 2014, Van Roy and Scheldeman, 2016a, 2016b, Johan et al., 2017).

Based on the experience from those studies, we established sensors mounted on a UAV to simultaneously measure the concentrations of $SO_2$ and $CO_2$ in order to calculate the FSC. The UAV can collect samples closer to the exhaust gas than airborne-based measurements. Waigaoqiao port in the Yangtze River Delta was selected as the study site. By using this measurement system, we analyzed 23 ship plumes and compared the results with the FSC of entering ships determined from fuel samples analyzed at certified laboratories. Through these experiments, we investigated and analyzed the emission process of $SO_2$ and $CO_2$ close to the funnel mouth of ships and design an accurate measurement of FSC.

## 2. Measurement

### 2.1 UAV

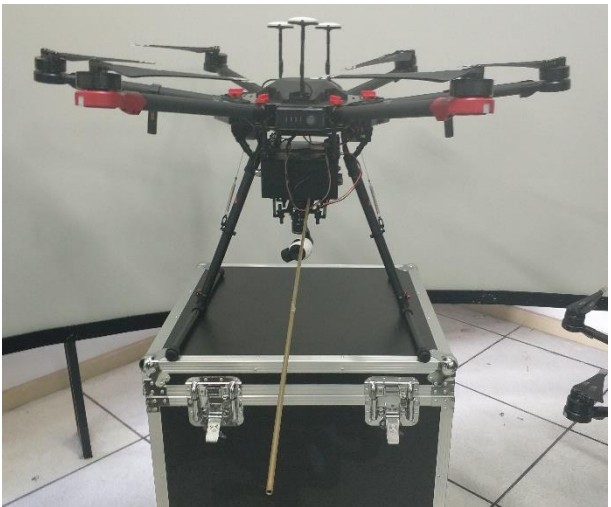

**Figure 1. Image of the modified UAV platform. The black box installed under the UAV is a pod which was designed and customized by us. It carries a gas pump (to collect the ship's exhaust gas), gas circuit, a filter (to remove water vapor), sensors for $SO_2$ and $CO_2$, a small motor (to provide energy for pumping), a camera, and communication modules.**

In the experiment, we used the MATRICE 600 UAV (SZ DJI Technology Co., Ltd.) with a few small modifications. We designed and customized a special pod, which was installed underneath the UAV, to carry sensors, communication circuit boards, gas circuit systems, and other modules, as shown in Fig. 1. After the successful assembly of the UAV platform, we first carried out preliminary experiments in the automatic engine room laboratory of Shanghai Maritime University. Through the preliminary test, we verified the stability and security of the whole UAV system. At the same time, it also allowed the UAV operator to practice how to operate the UAV for sampling close to the smoke stack. Fig. 2 shows a photograph of the process of collecting exhaust gas from near the smoke stack. The UAV can fly near the smoke for the collection and detection of exhaust gas. The detection information can be sent to the receiving end in real time. Table 1 presents the

parameters of the UAV. The weight of the pod is about 3 kg and the UAV can fly for about 25 min. Therefore, measurements can be taken from 1–2 ships using one set of batteries.

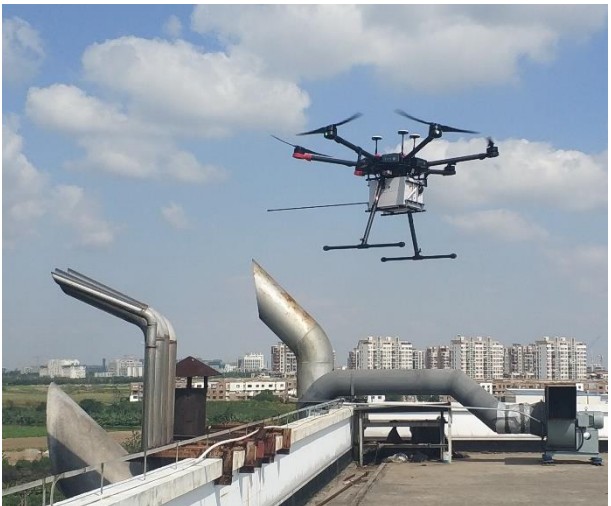

**Figure 2. UAV platform flying close to the smoke stack for collecting exhaust gas in the automatic engine room laboratory of Shanghai Maritime University.**

**Table 1. Parameters of the UAV**

| Parameter | Value |
|---|---|
| Symmetrical motor wheelbase | 1133 mm |
| Size | 1668 mm × 1518 mm × 727 mm |
| Weight | 9.5 kg |
| Recommended maximum take-off weight | 15.5 kg |
| Hovering accuracy(P-GPS) | Vertical: ±0.5 m, Horizontal: ±1.5 m |
| Maximum rotational angular velocity | pitch axis: 300°/s, Heading axis: 150°/s |
| Maximum pitch Angle | 25° |
| Maximum rising speed | 5 m/s |
| Maximum rate of descent | 3 m/s |
| Maximum sustained wind speed | 8 m/s |
| Maximum horizontal flight speed | 65 km/h (no wind environment) |
| Hover time | non-loaded:32 min,   load 6 kg:16 min |

**2.2 Sensors**

In the measurement process, the ship exhaust gas is pumped into the pod by the gas pump. After the filter removes the water vapor, the sensors react and the communication module sends the measurement results to the receiving end. The sensors

included instrumentation for both $SO_2$ and $CO_2$ measurements. These sensors were purchased from HH Feuerungstechnik GmbH, Germany.

For $SO_2$, the sensor is based on the electrochemical method. An electrochemical sensor determines the concentration of a gas via a redox reaction, producing an electrical signal proportional to the concentration of the gas. In previous measurements of ship exhaust gas, $SO_2$ sensors are mainly based on the UV-fluorescence method (Balzani et al., 2014, Beecken et al., 2014, Kattner et al., 2015, Johan et al., 2017), which is not appropriate for the UAV due to weight limitations. The $SO_2$ electrochemical sensor has the advantages of low power consumption, small size, light weight, and high precision. In addition, this type of sensor is capable of measuring $SO_2$ at a low ppb range (Hodgson et al., 1999). Therefore, we used the electrochemical sensor to measure $SO_2$ concentration. The measuring range of the sensor is 0–5 ppm, the resolution level is 0.001 ppm, response time ($t_{90}$) is less than 1 s, and the accuracy is ±0.25 ppm. $t_{90}$ is defined as the time it takes to reach 90% of the stable response after a step change in the sample concentration.

For $CO_2$, the sensor is based on the non-dispersive infrared analyzer method. This type of sensor is often used to measure the $CO_2$ concentration of ship exhaust gas (Balzani et al., 2014, Beecken et al., 2014, Kattner et al., 2015, Johan et al., 2017). An infrared beam passes through the sampling chamber, and each gas component in the sample absorbs infrared rays at a specific frequency. The concentration of the gas component is determined by measuring the infrared absorption at the corresponding frequency. The measuring range of the used sensor is 0–5000 ppm, resolution level is 1 ppm, response time ($t_{90}$) is less than 1 s, and its accuracy is ±50 ppm.

Sensor calibration is required when the equipment is used daily. The time interval for sensor calibration is three months or when the accumulated working time of the sensor exceeds 180 h. If either of these conditions is met, calibration will be carried out. The zero and full scales are usually calibrated by standard mixture gas. Before each mission, sensors are activated and residual gas in the airway is discharged by the gas pump.

## 3. Methods

### 3.1 Flight procedures

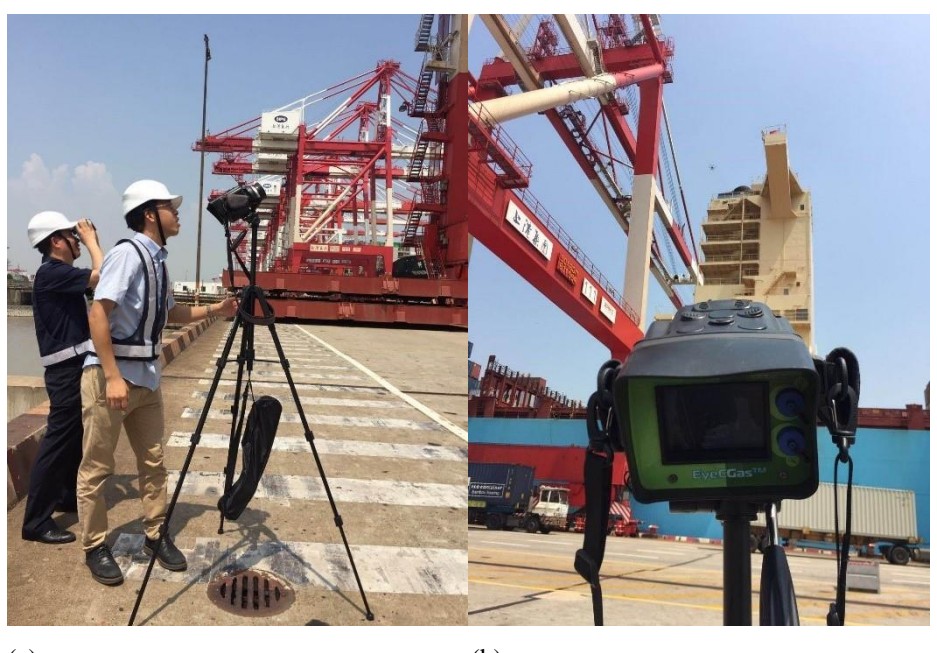

(a)                (b)

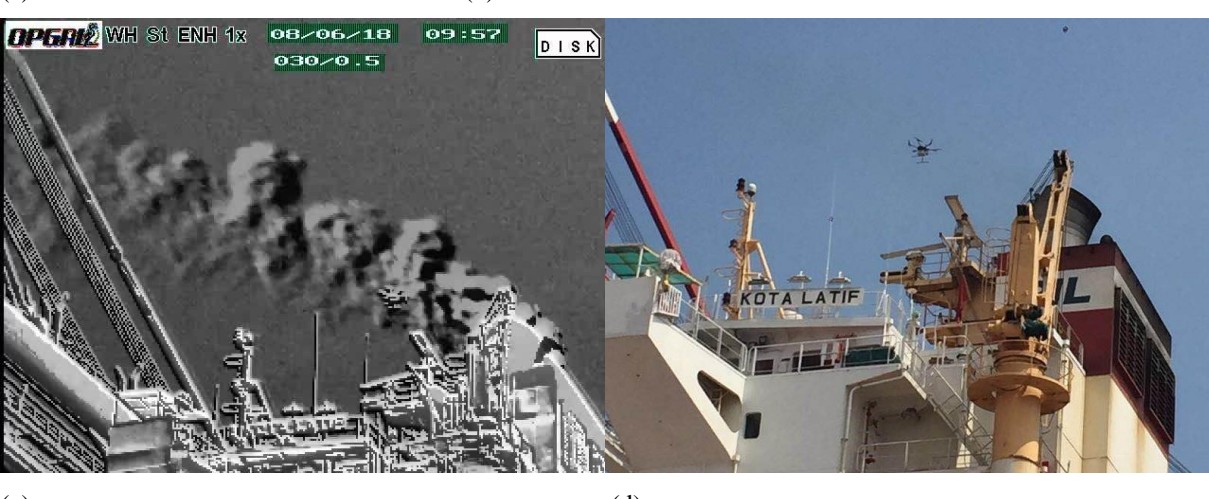

(c)                (d)

**Figure 3. Photographs showing the setup of the experiment. An infrared camera is set up for locating the smoke plume (a), (b). The target plume is imaged by the infrared camera (c). The UAV takes off towards the smoke plume (d).**

The preliminary positioning measurements of the ship smoke plume are as shown in Fig. 3. The UAV platform with sensors flew close to the funnel of ship, hovered for collecting exhaust gas, and then detection information was sent back. This

procedure is not without risk and a well-considered flight approach is recommendable. We summarize the experiment steps as follows:

1. Determine the position of the plume according to the wind speed, wind direction, height gauge, infrared camera, and other factors.

2. Check the equipment to ensure that the power is sufficient, the GPS signal is normal (it is recommended that the number of satellites is more than 13), the electrochemical sensor is activated, and the residual gas is discharged in the air path of the pod.

3. The UAV takes off vertically and rises to an altitude of 100 m (the first measurement point) for 3 min to determine the background value of $SO_2$ and $CO_2$. The take-off position is usually on the dock and is more than 50 m away from the ship's

smoke.

4. Fly the UAV towards the plume and hover to collect exhaust gas from about 10 m (the second measurement point) and 5 m (the third measurement point) away from the funnel for 5 min, respectively.

5. Lift the UAV and then return it to the starting point.

During the process, real-time observations of $SO_2$ and $CO_2$ were sent to receiving end. The operator adjusted the UAV's

position according to the observations to keep the sensors in the plume. Therefore, in general, the UAV confirmed the approximate location of the plume at a distance of 10 m, and then gradually approached the location of about 5 m for collection.

### 3.2 Calculation of FSC

When the UAV flew into the ship' plume, the peak areas of the $SO_2$ and $CO_2$ measurements were determined, and the

background was subtracted. The background value of $SO_2$ and $CO_2$ is obtained when the UAV hovers at the first measurement point. The peak values of $SO_2$ and $CO_2$ are determined when the UAV hovers at the second measurement point or the third measurement point (main observation point). In the calculation, the molecular weights of carbon and sulfur are 12 g mol$^{-1}$ and 32 g mol$^{-1}$, respectively, and the carbon mass percent in the fuel is 87$\pm$1.5% (Cooper et al., 2003). With the assumption that 100% of the sulfur and carbon contents of the fuel are emitted as $SO_2$ and $CO_2$, respectively, the FSC mass

percent can be expressed as follows:

$$FSC[\%] = \frac{S[kg]}{fuel[kg]} = \frac{SO_2[ppm] \cdot A(S)}{CO_2[ppm] \cdot A(C)} \cdot 87[\%] = 0.232 \frac{\int (SO_{2,peak} - SO_{2,bkg})dt[ppb]}{\int (CO_{2,peak} - CO_{2,bkg})dt[ppm]}[\%] \qquad (1)$$

where $A(S)$ is the atomic weight of sulfur and $A(C)$ the atomic weight of carbon. $SO_{2,peak}$, $SO_{2,bkg}$, $CO_{2,peak}$, and $CO_{2,bkg}$ are the peak and background values of $SO_2$ and $CO_2$, respectively. This calculation method is consistent with that described in the MEPC guidelines 184(59) and previous studies (Beecken et al., 2014, Kattner et al., 2015, Johan et al., 2017).

The response time of both sensors is less than 1s. Even if the sampling rates of the two sensors are set to be consistent, the two sensors cannot be completely synchronized. This makes it difficult to calculate the instantaneous ratio of $SO_2$ and $CO_2$. Our approach is that the sensor sends the average measurement value of the last 10 s to the receiver at an interval of 10 s.

Therefore, the interval of integration in Eq. (1) is 10 s. We found that taking the mean of measurements directly or at shorter intervals leads to too many narrow peaks in one measurement process. This makes it difficult to select the peak value, and the calculation results are unstable. At the same time, the interval should not be set too long, which will make the crest very inconspicuous or too flat. Therefore, we selected 10 s as the empirical parameter value after several experiments.

## 3.3 Uncertainties

Because measurements taken inside the ship plumes are analyzed relative to the background, offset errors can be neglected. Nevertheless, there are certain uncertainties in the estimation process of the FSC. They can be summed up as sensor uncertainty, measurement uncertainty, calculation uncertainty, exhaust uncertainty, and so on.

Regarding sensor uncertainty, the nonlinearity of the two sensors should be no more than ±1% and the linear error is negligible. It can be corrected through frequent calibrations with standard gases and gradually establishing a quality management system comprising sensor linearity, sensitivity, repeatability, hysteresis, resolution, stability, drift, and other attributes of the minimum requirements.

Measurement uncertainty is mainly attributable to inadequate sampling (the UAV did not fly into the plume). Moreover, shipborne antennae, dock facilities, and strong winds may cause interference in finding an appropriate sampling point and even lead to sampling failure. This uncertainty factor can lead to an incorrect estimation of the FSC. Therefore, we formulated the flight procedures as described in section 3.1.

Calculation uncertainty lies in selecting the background and peak values of $SO_2$ and $CO_2$. According to the law of error propagation (widely used in surveying, mapping, and statistics), the relationship between the deviation in the measurement values and that in the FSC can be obtained. The FSC calculation results are functions of independent observations $SO_{2,peak}$, $SO_{2,bkg}$, $CO_{2,peak}$, and $CO_{2,bkg}$ as in Eq. (1). The relationship between the observation error ($\Delta SO_{2,peak}$, $\Delta SO_{2,bkg}$, $\Delta CO_{2,peak}$, and $\Delta CO_{2,bkg}$) and function error ($\Delta FSC$) can be approximated using the full differential of the function as follows:

$$\Delta FSC = \frac{\partial f}{\partial SO_{2,peak}}\Delta SO_{2,peak} + \frac{\partial f}{\partial SO_{2,bkg}}\Delta SO_{2,bkg} + \frac{\partial f}{\partial CO_{2,peak}}\Delta CO_{2,peak} + \frac{\partial f}{\partial CO_{2,bkg}}\Delta CO_{2,bkg} \tag{2}$$

In our study, this deviation was generally in the order of hundreds of ppm, as explained in section 4.

Exhaust uncertainty arises because not all the sulfur in the fuel is emitted as $SO_2$, which is a systematic uncertainty. Preliminary studies showed that 1-19% of the sulfur in the fuel is emitted in other forms, possibly $SO_3$ or $SO_4$ (Schlager et al., 2006, Balzani Lööv et al., 2014). Hence, the assumption that all sulfur is emitted as $SO_2$ yields an underestimation of the true sulfur content in the fuel. Accordingly, this factor needs to be considered when setting the alarm threshold of the FSC.

In any case, these uncertainties will occur during the measurement process. After the establishment of flight procedures as mentioned in section 3.1 and selection process as in section 4, we observed that the deviation between the estimated value of FSC and true value of FSC was generally not more than 300 ppm. In addition, none of the monitored ships were fitted with exhaust cleaning equipment.

# 4. Results

## 4.1 Data treatment

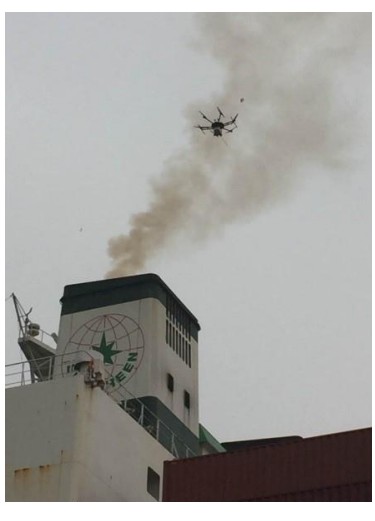 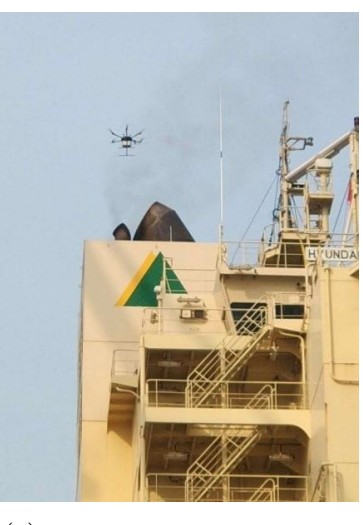

(a)                                    (b)

**Figure 4. Photographs showing the flight of the UAV during measurements. The UAV platform was flown close to the funnel of ship for collecting exhaust gas and detection at Waigaoqiao pier.**

Fig. 4 shows the UAV platform with sensors flying close to the ships plume. It hovered to collect exhaust gas, and detection information was subsequently sent back. Generally, changes in $SO_2$ and $CO_2$ observations can be divided into three stages: (1) The UAV took off and approached the ship funnel for about 3 min. The $SO_2$ and $CO_2$ observations were relatively low, and the background value was obtained in this stage. (2) The UAV was gradually flown to the plume center, and data were collected. Rapid increases in $SO_2$ and $CO_2$ concentrations, reaching their peaks, were observed, which took approximately 10–15 min. The peak data were obtained in this stage. (3) The UAV completed the gas collection and returned, which took about 5 min. Decreased $SO_2$ and $CO_2$ concentrations relative to the observation when the UAV was in the plume center were observed. Observed $SO_2$ and $CO_2$ values returned to background levels, but they were not used as background values. Residual gas in the airway needed to be discharged by the gas pump before the next collection.

Numerous measurements have been made in the Waigaoqiao wharf since January 2018. After the adjustment of various technical parameters and the accumulation of UAV flight experience, this method could provide accurate results. From August 2018 to January 2019, 23 plumes exhausted by ships have been detected. Fuel samples, which are considered as the true value of FSC, were taken and sent for laboratory chemical examination. Finally, the results of the UAV method were compared with those of the laboratory tests.

According to Eq. (1), if the observations of $SO_2$ and $CO_2$ values simultaneously reach their peaks, it is easier to select the background and peak value to calculate the FSC. However, the actual data collected are sometimes not ideal, and there is calculation uncertainty when selecting the background and peak values of $SO_2$ and $CO_2$. In previous studies, procedures for selecting background and peak values were not discussed in detail. As the number of experiments increased, we gradually

developed a selection process. In our experiment, observations of $SO_2$ and $CO_2$ in the receiving end were synchronized. Therefore, the background and peak values for $SO_2$ and $CO_2$ that we selected to calculate the FSC were observed at the same time point.

According to the flight record, the minimum values of $SO_2$ and $CO_2$ collected at the first measurement point are selected as the background values. There is generally greater uncertainty in selecting the peak values. The synchronous, stable, obvious, and maximal values in observations of $SO_2$ and $CO_2$ are selected as the peak values. The selection method is as follows:

1. The peak values in the observations of $SO_2$ and $CO_2$ are determined at the second and third measurement points, respectively.

2. The peak values at the full range of the $SO_2$ or $CO_2$ sensors are ruled out.

3. The peak values resulting from dramatic changes (for instance, if the change in $CO_2$ exceeded 500 ppm, or if the change in $SO_2$ exceeded 500 ppb) in continuous observations are ruled out, because these changes may have been related to sensor uncertainty, exhaust uncertainty, or unstable concentrations of $SO_2$ or $CO_2$ in the atmosphere.

4. The occurrence time of peak values in $SO_2$ and $CO_2$ are compared, and then the simultaneous peaks and almost simultaneous peaks (no more 20 s apart) are retained. If there is a small deviation between the time point of the peak values for $SO_2$ and $CO_2$, we select the time point at peak of $SO_2$. This will make the FSC value relatively larger than that of $CO_2$. As in Eq. (1), a higher $SO_2$ peak leads to a higher FSC estimate, while a higher $CO_2$ peak leads to a lower FSC estimate. As discussed in section 3.3, not all the sulfur in the fuel is emitted as $SO_2$, which will result in a lower estimate value. This selection allows the estimate to be relatively close to the true value.

5. After the above filtration, approximately 1–4 time points will be left as the selection points for peak values. The global maximum values are selected as peak values to calculate the FSC. The maximum values are likely to have been measured in the center of the ship's plume. At that location, the measurement value is relatively stable, and the probability of interference from other factors is lower.

## 4.2 FSC estimation

In our experience, using the above method can provide the FSC value that is closest to the real value in most cases. In a few cases, it may be suboptimal rather than optimal. However, the final deviation generally does not exceed 0.03% (m/m) at an FSC level of 0.035% (m/m) to 0.24% (m/m). To illustrate this selection method, six typical sets of plume measurement data for $SO_2$ and $CO_2$, marked as plumes 1–6, along with the time and serial number, are shown in Fig. 5. In addition, we made a distinction between good-and poor-quality data and rejected some plumes. Good-quality data for a plume meant that the peak values were obvious and easy to distinguish, whereas poor-quality data for a plume meant that the peak values were less obvious but still able to produce a result. When results could not be obtained, the plumes were rejected. An FSC of 0.01% (m/m) was used as the dividing line between plumes with high-sulfur and low-sulfur content samples.

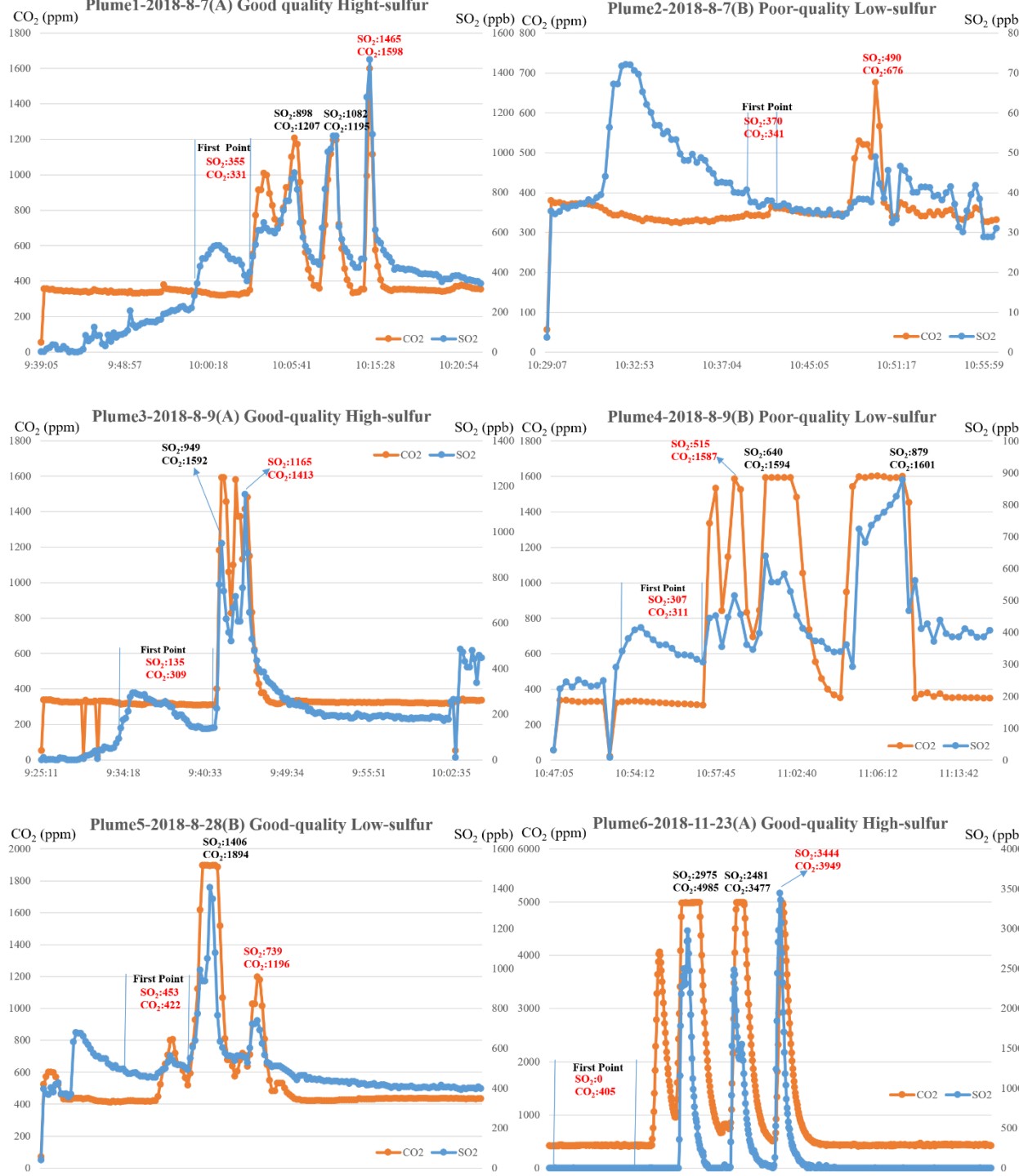

**Figure 5. Six sets of plume measurement data for SO₂ and CO₂, marked as plumes 1–6, along with the time and serial number. The**
**background and peak values of SO₂ and CO₂ were used to estimate the FSC. In each plume, the time range of the first monitoring**

**point is marked by two vertical lines. The selected background and peak values of SO₂ and CO₂ are written in red and alternative peak values are written in black.**

As shown in Fig. 5, the observations of plumes 1 and 3 simultaneously reached the peak value. However, these were multiple $SO_2$ and $CO_2$ peak values, and the global maximum peak values of $SO_2$ and $CO_2$ were selected. In plume 2, there was a peak for $SO_2$ at 10:32, but there was none for $CO_2$ at the same time. We used the data from the simultaneous peaks of $SO_2$ and $CO_2$ for the calculations. The observations of plumes 4 and 5 also simultaneously reached the peak value at multiple time points. However, at 11:02 and 11:07 in plumes 4 and 11:19 in plume 5, the $SO_2$ measurements reached the peak values, but the $CO_2$ measurements reached plateau levels above which they did not increase any further. Therefore, the data in this period were not used as peak values of the plumes. In plume 6, $CO_2$ measurements did not increase any further owing to the full range of the $CO_2$ sensor at 10:02 and 10:04. This happens in rare cases when the UAV is too close to the funnel (less than 5 m), and these data cannot be used as peak values. After the measurement of plume 5, the communication module was fault when we wanted to adjust sampling rate. We consequently replaced the communication protocol "HTTP protocol" with the "TCP/IP protocol". The main changes involved adjusting the data sampling rate from 10 to 2 s to make it easier to find the peak value (the sensors send the average measurement value of the last 10 s to the receiver at an interval of 2 s), and the sensors were consequently recalibrated by standard mixture gas. Therefore, the background values of plumes 1–5 were different from those of plume 6. Nonetheless, Eq. (1) was used to calculate the ratio of sulfur dioxide difference to carbon dioxide difference, and it therefore does not affect the final calculation results. In addition, when the FSC of the target ship is low, for example, when the fuel used is light diesel fuel, the $SO_2$ observation values were mostly 0. When this happened, according to our experience, the FSC was generally lower than 200 ppm, and the ship was likely to meet the emission requirements.

The background and peak values of $SO_2$ and $CO_2$ were selected from plumes 1–6, and the FSC was calculated according to Eq. (1). The comparison results of the estimated FSC values are presented in Table 2. The background value of $CO_2$ in plumes 1–4 exceeded 300 ppm, but the global background $CO_2$ was approximately 400 ppm. Meanwhile, the background value of $SO_2$ exceeded 400 ppb at some time. This was due to sensor calibration, which did not affect the final result. This kind of situation did not happen again after we recalibrated the sensors by standard mixture gas. In some cases, background values seemed to fluctuate greatly. This was mainly because the UAV took off from the dock, where multiple ships were berthed, and wind speeds were high. In addition, the drift or cross sensitivity in the sensors also may have caused interference. Therefore, we used the flight procedure given in section 3.1 and the selection method of peak values to minimize this impact. By comparing the results and deviations of the different calculated values, it can be seen that appropriately selecting the peak value is important. In general, the optimal value can be selected using the selection method with the exception of plume 1. However, the deviation is not large.

**Table 2 Comparison and verification of the estimated and true values of FSC. We present the selected background and peak values of SO₂ and CO₂ and alternative peak values (mentioned in Figure 5). The FSC results and deviations of these different values are also listed for comparison purposes. They are distinguished as follows in the column titled "Selected": the selected peak values are marked " √ " indicates the selected peak values, and "×" indicates alternative peak values (which is not selected as the calculated value in the final result of FSC).**

| ID | Plume ID | Selected | SO₂ (ppb) | | CO₂ (ppm) | | Estimated value of FSC (ppm) | True value of FSC (ppm) | Deviation (ppm) |
|---|---|---|---|---|---|---|---|---|---|
| | | | Bkg | Peak | Bkg | Peak | | | |
| 1 | | √ | | 1465 | | 1598 | 2033 | | 110 |
| 2 | Plume1 | × | 355 | 1082 | 331 | 1195 | 1952 | 1923 | 29 |
| 3 | | × | | 898 | | 1207 | 1438 | | -485 |
| 4 | Plume2 | √ | 370 | 490 | 341 | 676 | 831 | 954 | -123 |
| 5 | Plume3 | × | 135 | 949 | 309 | 1592 | 1472 | 2113 | -641 |
| 6 | | √ | | 1165 | | 1413 | 2164 | | 51 |
| 7 | | √ | | 515 | | 1587 | 378 | | -18 |
| 8 | Plume4 | × | 307 | 640 | 311 | 1594 | 602 | 396 | 206 |
| 9 | | × | | 879 | | 1601 | 1029 | | 633 |
| 9 | Plume5 | √ | 453 | 739 | 422 | 1196 | 857 | 868 | -11 |
| 10 | | × | | 1406 | | 1894 | 1502 | | 634 |
| 11 | | √ | | 3444 | | 3949 | 2255 | | -132 |
| 12 | Plume6 | × | 0 | 2481 | 405 | 3477 | 1874 | 2387 | -513 |
| 13 | | × | | 2975 | | 4985 | 1507 | | -880 |

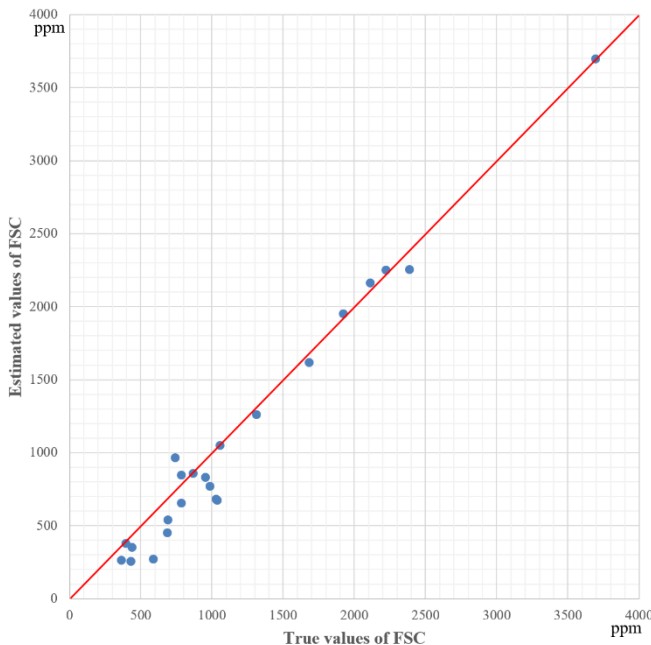

**Figure 6. Comparison between the true values of FSC (x-axis) against the estimated values of FSC (y-axis) of 23 times measurement.**

As shown in Fig. 6, the FSC in our experiments was mainly at a level of 0.035% (m/m) to 0.24% (m/m). There was one measurement of 0.37% (m/m), especially. However, it is not enough to illustrate the deviation at the level of 0.24% (m/m) to 0.37% (m/m), because deviations of FSC are not the same at different FSC levels. Overall, the estimated FSC is smaller than the true value in many cases. This could be due to the exhaust uncertainty that not all the sulfur in the fuel is emitted as $SO_2$.

In our experiments, this uncertainty factor led to low FSC estimation results, and the deviation was generally not more than 200 ppm. This prediction is based on the fact that several measurements of some plumes were taken at particular times. Similar calculation results for FSC were obtained, but they were all less than the real value of 100–200 ppm. This tendency of underestimation has also been found in previous studies (Johan, R et al. 2017).

Finally, the deviation of the estimated FSC value calculated using the proposed method was within 300 ppm (0.03% (m/m)),
although there was some uncertainty. Considering the uncertainties listed in section 3.3, the proposed method provides accurate results.

## 5. Conclusions

In this study, we performed close monitoring of ship smoke plumes using UAV. Observation data of $SO_2$ and $CO_2$ were collected at close range (5–10 m) of ship funnel mouths. The estimated results were compared with the FSC values
determined at certified laboratories. In general, the deviation of the estimated FSC value was within 0.03% (m/m) at an FSC level of 0.035% (m/m) to 0.24% (m/m). Because not all the sulfur in the fuel is emitted as $SO_2$, the estimated FSC is smaller than true value in many cases. Therefore, if the maritime department wants to take the estimated value as the basis for the preliminary judgment regarding whether the ship exceeds the emission standard, it needs to set an appropriate threshold and a confidence interval.

At present, the FSC limit in China's emission control requirements is 0.5% (m/m), and that for ECAs is 0.1% (m/m). The proposed method can be used for monitoring of ECAs for compliance with FSC standards. However, after more than one year of testing and experiment, we found that there are still many issues that remain to be resolved:

1. In about 10% of the cases, the UAV did not measure the effective background value and peak value. This is mainly caused by the UAV missing the plume during its flight. Therefore, effective methods for finding and navigating to plumes using
real-time sensor feeds need to be explored.

2. In about 10% of the cases, the absolute error was more than 0.03% (m/m), and even more than 0.05% (m/m) in rare cases. Unstable concentrations of $SO_2$ or $CO_2$ in the atmosphere just before the measurement may cause such errors. Furthermore, uncertainties, such as sensor uncertainty, measurement uncertainty, calculation uncertainty, and exhaust uncertainty, may hinder accurate measurement. Poor-quality data or rejected plumes may result from these situations, i.e., unstable
concentrations of $SO_2$ or $CO_2$ and uncertainties.

3. Currently, the pod can only carry two sensors. In subsequent tests, we will modify the pod to carry more sensors. The use of different types of UAVs also needs to be evaluated. In addition, our experiments mainly involved the monitoring of berthing ships, and experiments on ships at sea are needed in the future.

*Data availability.* Requests for data sets and materials please address to Fan Zhou (fanzhou_cv@163.com).

*Author contributions.* FZ designed the study, analyzed the experimental data and authored the article. SP, WC and XN contributed to the experiments. BA provided constructive comments on this research.

*Competing interests.* The authors declare that they have no conflict of interest.

*Acknowledgments.* We thank Editage [www.editage.cn] for English language editing. We thank Duyzer, J. and one anonymous reviewer for reviewing this paper. We thank Folkert Boersma for serving as editor.

*Financial support.* This research has been supported by the National Natural Science Foundation of China (grant No. 41701523) and the Special Development Fund for China (Shanghai) Pilot Free Trade Zone.

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
