# Peer review of "Monitoring of compliance with fuel sulfur content regulations through UAV measurements of ship emissions"

_Atmospheric Measurement Techniques, 2019_

## Referee Comment (RC1) · Anonymous Referee #1 · 18 Apr 2019

Ship emission is a broadly investigated topic due to its significant effect on climate, air quality and hence people's health and welfare. It is especially important to monitor in Emission Controlled Areas (ECAs) whether the ships comply with the regulations or not. In this context the authors did an important and relevant research that might have scientific significance, which is, unfortunately, not reflected in current version of the manuscript.

Even the title is sloppy: "High-precision monitoring of compliance with fuel sulfur content through UAV measurements of ship emissions" Should be rather: "High-precision monitoring of compliance with fuel sulfur content regulations/limitation/standards

[Figure]

through UAV measurements of ship emissions"

1. Introduction

The literature review is poor. For example: Page1, line20: Year of citation is 2005. Add current data here since anthropocentric $SO_2$ emission might changed significantly during 14 years. Pg1, ln25: "some regulation went into effect" - needed to be rephased.

Pg2, ln4: "To cope with..." ???

The overview of techniques (Pg2, ln15-ln26) is incorrect. The optical methods (LIDAR, UV cam, DOAS) can measure only the $SO_2$ emission rate. For emission factor calculation simultaneous $CO_2$ emission rate measurement is needed preferable on the same parcel of the plume. It can be implemented by open path FTIR technique but it is quite challenging. Another solution could be to model the $CO_2$ emission rate based on the ship's technical properties and sailing characteristics. Then the $SO_2$ or S emission factor (EF) as well as the fuel sulphur content (FSC) can be calculated.

Pg2, ln25: What is the effect of NOx sensor on FSC measurement?

In addition, several UAV application have been done before that must be mentioned here. As a summary; a new and more precise Introduction is needed.

2. Measurement:

Avoid mini-sniffer term. Sniffing technique supposes an airflow through the analyzer. In this context the $CO_2$ analyzer can sniff but the $SO_2$ sensor definitely cant. Better to use sensing or electrochemical sensing terms.

This section is also incomplete. What sensors were used? Manufacturers, types, characteristics? The description of the sniffing technique (Pg5) is poor and incomplete. For example, the proper handling of the water vapor interference with $CO_2$ measurements is crucial. How did the authors handle that?

Ln13: "calibrated 3 month or 180 working hours apart" ??? - why is this big difference?

I miss evidences of laboratory experiments where the sensors were calibrated and tested, effects of environmental factors (temperature, humidity) were investigated as well as interference of other components (water vapor) was checked.

4. Results

The authors claimed that they measured 20 ships. Why only 6 plumes were presented here? It is not clear how the authors accepted or discarded results. What were the main steps of the consideration? On the other hands, the main strength of the work that the authors compared their plume measurements with the chemical analysis of the fuel. The authors did not mention the biggest challenge of the technique, namely how can we synchronize the time variations of two different measurements ($SO_2$ and $CO_2$) in order to calculate their ratio. In case of broader plume parcels (and thus longer measurement time) during conventional sniffing technique the uncertainty of the integrals are negligible, so more or less exact ratio can be calculated. On the other hands, the sensing technique provides narrow peaks of both components where the uncertainty of the integral is significant.

How can the authors describe the differences in the time variation of the two components (see the figures on Pg10)? In Plume 5 and Plume 6 the $CO_2$ sensor was saturated at different concentrations (1900 vs 5000 ppm). What was the reason?

As a summary; although the authors did significant efforts and they have some good results this work is still not ready for publication. I encourage the authors to consider my comments and fix the weaknesses of the paper. A completed and corrected description of this work would be worth to publish in AMT.

---

## Referee Comment (RC2) · Anonymous Referee #2 · 30 Apr 2019

Review: Atmos. Meas. Tech Discuss. (amt-2019-29) High-precision Monitoring of compliance with fuel sulfur content through UAV measurement of ship emissions. By Fan Zhou et al. General comments Title: Without trying to be negative I would suggest leaving out the words "High-precision". Abstract: As in the title I would suggest leaving out " high precision" in the last sentence. I would also mention the range of sulfur contents that were encountered in the study i.e. how many non-conformities were encountered. And I would like to mention more explicitly that the deviation of the estimated value for +FSC is less than 0.03% (m/m) at a level of 0.04 % to 0.24 % FSC. Note that in ECA areas, with a limit of 0.1%, an uncertainty of 0.03% is not very good. I would also suggest mentioning that in all cases the estimated FSC was always lower

than the actual FSC derived from samples taken on board. This is an important aspect with a strong impact on the usefulness of the method in SECA areas with a 0.1% limit value. Paper: This could be a very useful paper with lots of detail. Especially the level of detail is useful since this is an area with a lot of development and sharing of these new results could very helpful to other scientists. I provide some comments that could help to make the paper a bit clearer in some areas. See my specific comments below. Figure 1: I am not familiar with UAVs and in a first glance I thought the black box mentioned in the text was the large flight case black box below the drone. Page 3 line 16. Not everybody may be familiar with the word "Pod" Page 4 last sentence: electrochemistry method. Electrochemical method? Page 5 line 12-13. These sentences are rather unclear. What is meant with 180 working hours apart? Each 180 working hours? It is not entirely clear what the actual accuracy is if it is 1% full scale. Page 7 line 16: correction should be corrected. Gradually establishing a quality management system.…. Is rather vague what is meant. Please rephrase. Page 7 line 22. Here 200 ppm is mentioned where in other places in the text 0.03 % (300 ppm) is mentioned. This should be explained or there should rather be only one number. Same place: the deviations mentioned in Balzani et al. (2014) were determined at FSC of 1%. It is not clear whether these deviations are still the same at 0.1% FSC. They could be lower at 0.1% FSC content. The authors should mention that or provide more information (which would be useful) Page 7 last paragraph. To me it is not clear how errors in determination of the peak height is propagated in the total error and it is not clear how this is done. The error of 300 ppm is (it seems) related to the comparison with the on-board samples. And not from error propagation analysis as far as I can tell. It would be nice to show the error propagation numbers as well and see how well these two approaches match. In general, I think that the uncertainty discussion could be more quantitative.

Page 9 line 16: "this makes the FSC value relatively larger than that of CO2". It is not clear what is meant here. Page 9 line 6: were synchronized is rather vague. Please explain Page 9 In general, the data treatment is unclear to me. Why are peak values taken to compare SO2 and CO2? Or is it the surface area? The S-content may be

derived from any set of concentrations. Taking the pea area ins just of way of aver-aging. It seems to me now that the peak position and its height is depending on the performances of the sensors (especially response time) and the accidental position in the plume. This could lead to uncertainties especially if the peak height only is used. This should be explained better. Especially the "approach" could be elaborated more. Sometimes I am in doubt whether peak means the highest point in the concentration or the peak area. Page 9 in general: what exactly is "selected". This should be made clear. Now it seems a bit arbitrary. Of course, full range values are not used. But what are dramatic changes? Would be useful to explain. Page 9 line 21: 300 ppm at what level?? Page 10: Figure 5. Sometimes background values of SO2 are 400 ppb? That is very high. Why not subtract the background? Also in plume 6 the background seems to fluctuate very much. makes interpretation of peaks uncertain. please discuss, Page 11 table. Why is not a graph provided? Such as true value (x-axis) against estimated value (y-axis). Then also a correlation coefficient could be calculated. Also a good measure of quality. In general: The results section could improve in clarity if some structure was used: data treatment; FSC observed etc. For example, the issues with sampling rate etc. (page 1 top) are perhaps important but mixed here with the results. To increase clarity this could be treated separately

Conclusions High precision is not reasonable to state in view of the rather large under-estimations. Page 12: in Conclusions something might be said on the effect of SO3 and SO4

Specific comments: I am not a native speaker, but the English seems fine with me in general. Some specific text could be altered: - on ships the "chimney" is often called the "funnel" - "ship" is normally "vessel". - Culled is not a word that is often used - Page 8 line 3: English: none of the monitored ships were fitted with exhaust cleaning equipment

Please also note the supplement to this comment:

https://www.atmos-meas-tech-discuss.net/amt-2019-29/amt-2019-29-RC2-supplement.pdf

---

## Author Comment (AC1) · 15 May 2019

Answer to Referee #1

We would like to thank Referee #1 for his/her positive and constructive comments and suggestions. We have studied comments carefully and made corrections, which we hope meet with approval. Comments and responses are listed as follows. In order to facilitate the reference to the questions and proposed changes, we use the following color coding:

Color coding:

**Referee comment**

Our answer

Proposed change in manuscript

**The title is sloppy: "High-precision monitoring of compliance with fuel sulfur content through UAV measurements of ship emissions" Should be rather: "High-precision monitoring of compliance with fuel sulfur content regulations/limitation/standards through UAV measurements of ship emissions"**

Considering the content of the manuscript, we guess the word "regulations" is more appropriate. At the same time, Referee #2 suggest leaving out the word "High-precision" because "in ECA areas, with a limit of 0.1%, an uncertainty of 0.03% is not very good". In addition, with the consideration that the precision will improve further, with the progress of technology and method. "High-precision" seems not appropriate, we changed the title as:

"Monitoring of compliance with fuel sulfur content regulations through UAV measurements of ship emissions".

**1. Introduction**
**The literature review is poor.**

The suggestion is very important. We have rewritten this part carefully. The major changes are listed below.

**Page1, line20: Year of citation is 2005. Add current data here since anthropocentric SO$_2$ emission might change significantly during 14 years.**

This paragraph has been rewritten.

Estimations show that ships contribute 4-9% of global SO$_2$ emissions and 15% of NO$_x$ (Eyring et al., 2010). According to the United Nations Conference on Trade And Development (UNCTAD, 2017), the volume of the world's seaborne trade grew by 66% between 2000 and 2015. As global commerce expands, ocean-going ships consume more fuels, generally low-quality residual fuels containing high concentrations of sulfur and heavy metals (Lack et al., 2011). From the viewpoint of spatial distribution, the highest emissions of SO$_2$ per unit area occur in the eastern and southern China seas, sea areas in south-eastern and southern Asia, Red Sea, Mediterranean Sea, North Atlantic

near the European coast, Gulf of Mexico and Caribbean Sea, and along the western coast of North America. (Johansson et al., 2017). Ship-emitted pollutants influence air quality, human health, and climate. They not only affect the air quality in coastal areas but even influence the inland areas hundreds of kilometers away from the emission sources (Liu et al., 2016).

Reference:
[1] Eyring, V., Isaksen, I. S., Berntsen, T., Collins, W. J., Corbett, J. J., Endresen, O., Grainger, R. G., Moldanova, J., Schlager, H., and Stevenson, D. S.: Transport impacts on atmosphere and climate: Shipping, Atmos. Environ., 44, 4735–4771, https://doi.org/10.1016/j.atmosenv.2009.04.059, 2010.
[2] UNCTAD: World seaborne trade by types of cargo and by group of economies, annual, United Nations Conference on Trade and Development, available at: https://unctadstat.unctad.org/wds/TableViewer/tableView.aspx?ReportId=32363, last access: 5 March 2017.
[3] Lack, D. A., Cappa, C. D., Langridge, J., Bahreini, R., Buffaloe, G., Brock, C., Cerully, K., Coffman, D., Hayden, K., Holloway, J., Lerner, B., Massoli, P., Li, S.-M., McLaren, R., Middle-brook, A. M., Moore, R., Nenes, A., Nuaaman, I., Onasch, T. B., Peischl, J., Perring, A., Quinn, P. K., Ryerson, T., Schwartz, J. P., Spackman, R., Wofsy, S. C., Worsnop, D., Xiang, B., and Williams, E.: Impact of Fuel Quality Regulation and Speed Reductions on Shipping Emissions: Implications for Climate and Air Quality, Environ. Sci. Technol., 45, 9052-9060, https://doi.org/10.1021/es2013424, 2011.
[4] Johansson, L., Jalkanen, J. P., and Kukkonen, J.: Global assessment of shipping emissions in 2015 on a high spatial and temporal resolution, Atmos. Environ., 167, 403-415, https://doi.org/10.1016/j.atmosenv.2017.08.042, 2017.
[5] Liu, H., Fu, M., Jin, X., Shang, Y., Shindell, D., Faluvegi, G., Shindell, C., and He, K.: Health and climate impacts of ocean-going vessels in East Asia, Nat. Clim. Change., 6, 1037-1041, 10.1038/nclimate3083, 2016.

**Pg1, ln25: "some regulation went into effect" - needed to be rephased.**

These sentences have been rewritten.

In 2005, some regulations went into effect after being received by appropriate laws of the signatory states (at the European level it was received with the directives 1999/32/EC, 1999, and 2005/33/EC, 2005), and introduces limits to marine fuel sulfur content and engine performance to reduce $SO_x$ and $NO_x$ emissions. Further amendments to Annex VI were adopted in 2008 and entered into force in 2010.

Reference:
[1] Directive 1999/32/EC: Official Journal of the European Union, L 121, p. 13, 26 April 1999.
[2] Directive 2005/33/EC: Official Journal of the European Union, L 191, p. 59, 22 July 2005.

**Pg2, ln4: "To cope with..." ???**

This sentence has been rewritten.

In order to reduce the air pollution caused by ship emissions, the Atmospheric Pollution Prevention and Control Law of the People's Republic of China was promulgated in 2015 (Standing Committee of the National People's Congress, 2015).

Reference:
[1] Standing Committee of the National People's Congress, Atmospheric Pollution Prevention and Control Law of the People's Republic of China, 2015.

**The overview of techniques (Pg2, ln15-ln26) is incorrect. The optical methods (LIDAR, UV cam, DOAS) can measure only the $SO_2$ emission rate. For emission factor calculation simultaneous $CO_2$ emission rate measurement is needed preferable on the same parcel of the plume. It can be implemented by open path FTIR technique but it is quite challenging. Another solution could be to model the $CO_2$ emission rate based on the ship's technical properties and sailing characteristics. Then the $SO_2$ or S emission factor (EF) as well as the fuel sulphur content (FSC) can be calculated.**

In the manuscript, we only discuss that the optical methods can be used to measure ship emissions (do not indicate that they could measure the $CO_2$). We think this part was not clear enough, so we have rewritten it combined with the suggestion.

Optical methods analyze variations in light properties after interactions with the exhaust plume, and the local wind field before determining the $SO_2$ emission rate is observed. The simultaneous measurement of $CO_2$ and $SO_2$ emissions on a routine basis is unrealistic at present. Thus, the amount of fuel burned at the time of measurement is unknown and has to be estimated via modeling for calculating the FSC. For instance, the model STEAM (ship traffic emission assessment model), developed by the Finnish Meteorological Institute (Jalkanen et al., 2009) was used in the research for estimating FSC by Balzani Lööv et al. (2014). In addition, using the ratio of $SO_2$ and $NO_2$ measured via DOAS in the ship plume can be used as an indicator of FSC (Johan, R et al. 2017, Cheng, Y et al, 2019).

Reference:
[1] Balzani Lööv, J. M., Alfoldy, B., Gast, L. F. L., Hjorth, J., Lagler, F., Mellqvist, J., Beecken, J., Berg, N., Duyzer, J., Westrate, H., Swart, D. P. J., Berkhout, A. J. C., Jalkanen, J.-P., Prata, A. J., vander Hoff, G. R., and Borowiak, A.: Field test of available methods to measure remotely $SO_x$ and $NO_x$ emissions from ships, Atmos. Meas. Tech., 7, 2597–2613, doi:10.5194/amt-7-2597-2014, 2014.
[2] Cheng, Y., Wang, S., Zhu, J., Guo, Y., Zhang, R., Liu, Y., Zhang, Y., Yu, Q., Ma, W.,

and Zhou, B.: Surveillance of $SO_2$ and $NO_2$ from ship emissions by MAX-DOAS measurements and implication to compliance of fuel sulfur content, Atmos. Chem. Phys. Discuss., https://doi.org/10.5194/acp-2019-369, in review, 2019.
[3] Johan, R., Conde, V., Beecken, Jörg and Ekholm, J.: Certification of an aircraft and airborne surveillance of fuel sulfur content in ships at the SECA border, CompMon (https://compmon.eu/), 2017.
[4] Jalkanen, J.-P., Brink, A., Kalli, J., Pettersson, H., Kukkonen, J.,and Stipa, T.: A modelling system for the exhaust emissions of marine traffic and its application in the Baltic Sea area, Atmos. Chem. Phys., 9, 9209–9223, doi:10.5194/acp-9-9209-2009, 2009.

**Pg2, ln25: What is the effect of $NO_x$ sensor on FSC measurement?**

The $SO_2$ analyzer (fluorescence) response has cross sensitivity to NO. The supplementary explanation is given in the manuscript.

The "sniffing" method is based on simultaneous measurement of elevated $SO_2$ and $CO_2$ concentrations in the exhaust plume from the target ship and comparing them with the background. The measurement of $CO_2$ allows for relating the measurement of $SO_2$ to the amount of fuel burned at a given time, thus enabling the calculation of FSC directly. The concentration of $SO_2$ in plumes was generally measured using UV fluorescence or electrochemical sensors, and $CO_2$ was measured using a non-dispersive infrared analyzer (NDIR) or cavity ring down spectrometer (CRDS). The advantage of the "sniffing" method is that it offers more accuracy estimation for FSC. However, the instrument must be placed in the plume exhausted by the target ship. In some studies (Van Roy and Scheldeman, 2016a, 2016b), the "sniffing" method offers a measurement accuracy between 0.1–0.2% (m/m) FSC, which can be further increased up to 0.05–0.1% (m/m) FSC if combined with an additional $NO_x$ sensor. This is because the response of $SO_2$ analyzers (fluorescence) has cross sensitivity to NO. Deviations are not the same at different FSC levels, with an estimated relative uncertainty of 20% (m/m) for ships with 1% (m/m) FSC and a relative uncertainty of 50–100% at 0.1% (m/m) FSC. Balzani Lööv et al. (2014) obtained the following FSC measurements based on the "sniffer" principle: 0.86±0.23% (m/m) from land, 1.2±0.15% (m/m) from an on-board stack, and 1.13±0.18% (m/m) from a mobile platform. There was a 6% relative uncertainty for an FSC of 1% (m/m) but a 60% relative uncertainty for an FSC of 0.1% (m/m).

Reference:
[1] Balzani Lööv, J. M., Alfoldy, B., Gast, L. F. L., Hjorth, J., Lagler, F., Mellqvist, J., Beecken, J., Berg, N., Duyzer, J., Westrate, H., Swart, D. P. J., Berkhout, A. J. C., Jalkanen, J.-P., Prata, A. J., vander Hoff, G. R., and Borowiak, A.: Field test of available methods to measure remotely SOx and NOx emissions from ships, Atmos. Meas. Tech., 7, 2597–2613, doi:10.5194/amt-7-2597-2014, 2014.

[2] Van Roy, W. and Scheldeman, K.: Results MARPOL Annex VI Monitoring Report Belgian Sniffer Campaign 2016, CompMon (https://compmon.eu/), 2016a.
[3] Van Roy, W. and Scheldeman, K.: Best Practices Airborne MARPOL Annex VI Monitoring, CompMon (https://compmon.eu/), 2016b.

**In addition, several UAV applications have been done before that must be mentioned here.**

After reviewing relevant literatures, we found that there are some UAVs used to measure greenhouse gases and volcanic eruptions. Only one paper has been found on the measurement of ship emissions. We also have carried on the supplementary discussion in the manuscript.

Ship emission measurements can be divided into land-based (Kattner et al., 2015, Yang et al., 2016), marine-based (Cappa et al., 2014), airborne-based (Beecken et al., 2014, Aliabadi et al., 2016), satellite-based (Ding et al., 2018) and Unmanned Aerial Vehicle (UAV)-based (Villa et al., 2019) according to different platforms.
**…**
UAV-based measurements have gradually increased in the research regarding the atmosphere (Mori et al., 2016, Malaver Rojas et al., 2015). However, to date, there are relatively few applications of these measurements in ship emissions.

Reference:
[1] Aliabadi, A. A., Thomas, J. L., Herber, A. B., Staebler, R. M., Leaitch, W. R., Schulz, H., Law, K. S., Marelle, L., Burkart, J., Willis, M. D., Bozem, H., Hoor, P. M., Köllner, F., Schneider, J., Levasseur, M., and Abbatt, J. P. D.: Ship emissions measurement in the Arctic by plume intercepts of the Canadian Coast Guard icebreaker Amundsen from the Polar 6 aircraft platform, Atmos. Chem. Phys., 16, 7899–7916, doi:10.5194/acp-16-7899-2016, 2016.
[2] Beecken, J., Mellqvist, J., Salo, K., Ekholm, J., and Jalkanen, J.P.: Airborne emission measurements of $SO_2$, $NO_x$ and particles from individual ships using a sniffer technique, Atmos. Meas. Tech., 7, 1957–1968, doi:10.5194/amt-7-1957-2014, 2014.
[3] Cappa, C. D., Williams, E. J., Lack, D. A., Buffaloe, G. M., Coffman, D., Hayden, K. L., Herndon, S. C., Lerner, B. M., Li, S.M., Massoli, P., McLaren, R., Nuaaman, I., Onasch, T. B., and Quinn, P. K.: A case study into the measurement of ship emissions from plume intercepts of the NOAA ship Miller Freeman, Atmos. Chem. Phys., 14, 1337–1352, doi:10.5194/acp-14-1337-2014, 2014.
[4] Ding, J., van der A, R. J., Mijling, B., Jalkanen,J.-P.,Johansson,L.,and Levelt,P.F.: Maritime NOx emissions over Chinese seas derived from satellite observations. Geophysical Research Letters, 45, 2031–2037, doi:10.1002/2017GL076788, 2018.
[5] Kattner, L., Mathieu-Üffing, B., Burrows, J. P., Richter, A., Schmolke, S., Seyler, A., and Wittrock, F.: Monitoring compliance with sulfur content regulations of shipping

fuel by in situ measurements of ship emissions, Atmos. Chem. Phys., 15, 10087–10092, doi:10.5194/acp-15-10087-2015, 2015.

[6] Villa, T. F., Brown, R. A., Jayaratne, E. R., Gonzalez, L. F., Morawska, L., and Ristovski, Z. D.: Characterization of the particle emission from a ship operating at sea using an unmanned aerial vehicle, Atmos. Meas. Tech., 12, 691-702, https://doi.org/10.5194/amt-12-691-2019, 2019.

[7] Yang, M., Bell, T. G., Hopkins, F. E., and Smyth, T. J.: Attribution of atmospheric sulfur dioxide over the English Channel to dimethyl sulfide and changing ship emissions, Atmos. Chem. Phys., 16,4771–4783, https://doi.org/10.5194/acp-16-4771-2016, 2016.

**As a summary; a new and more precise Introduction is needed.**

We have tried our best to rewrite the introduction.

**2. Measurement:**
**Avoid mini-sniffer term. Sniffing technique supposes an airflow through the analyzer. In this context the $CO_2$ analyzer can sniff but the $SO_2$ sensor definitely cant. Better to use sensing or electrochemical sensing terms.**

The term of "mini-sniffer" in the manuscripts has been revised.

**This section is also incomplete. What sensors were used? Manufacturers, types, characteristics? The description of the sniffing technique (Pg5) is poor and incomplete. For example, the proper handling of the water vapor interference with $CO_2$ measurements is crucial. How did the authors handle that?**

The sensors are commercially available. This information has been supplemented.

In the measurement process, the ship exhaust is pumped into the pod by the gas pump. After the filter removes the water vapor, the sensors react and the communication module sends the measurement results to the receiving end. The sensors included instrumentation for both $SO_2$ and $CO_2$ measurements. These sensors were purchased from HH Feuerungstechnik GmbH, Germany.

In addition, we supplement the description in figure 1.

[Figure]

**Figure 1. Image of the modified UAV platform. The black box installed under the UAV is a pod which was designed and customized by us. It carries a gas pump (to collect the ship's exhaust), gas circuit, a filter (to remove water vapor), sensors for $SO_2$ and $CO_2$, a small motor (to provide energy for pumping), a camera, and communication modules.**

In the experiment, we used the MATRICE 600 UAV (SZ DJI Technology Co., Ltd.), and modified it. We designed and customized a special pod, which was installed underneath the UAV, to carry sensors, communication circuit boards, gas circuit systems, and other modules, as shown in Fig.1.

**Ln13: "calibrated 3 month or 180 working hours apart" ??? - why is this big difference?**

This sentence has been rewritten.

Sensor calibration is required when the equipment is used daily. The time interval for sensor calibration is three months or when the accumulated working time of the sensor exceeds 180 h. If either of these conditions is met, calibration will be carried out.

**I miss evidences of laboratory experiments where the sensors were calibrated and tested, effects of environmental factors (temperature, humidity) were investigated as well as interference of other components (water vapor) was checked.**

In the laboratory experiment, we mainly test the stability and safety of the whole UAV system as well as communication modules. At the same time, it also allows the UAV operator to practice how to operate the UAV for sampling close to the smoke stack. There is a risk of getting too close, the operator needs to practice.
We used commercial sensors. We chose the type of sensors according to the need of experiment. We have added some details (sources of sensors, technical parameters, calibration, filters, etc) about the whole pod as mentioned above. We hope this will make the readers more familiar with our work.

In the experiment, we used the MATRICE 600 UAV (SZ DJI Technology Co., Ltd.), and modified it. We designed and customized a special pod, which was installed underneath the UAV, to carry sensors, communication circuit boards, gas circuit systems, and other modules, as shown in Fig.1. After the successful assembly of the UAV platform, we first carried out preliminary experiments in the automatic engine room laboratory of Shanghai Maritime University. Through the preliminary test, we verified the stability and security of the whole UAV system. At the same time, it also allowed the UAV operator to practice how to operate the UAV for sampling close to the smoke stack.

**4. Results**
**The authors claimed that they measured 20 ships. Why only 6 plumes were presented here? It is not clear how the authors accepted or discarded results. What were the main steps of the consideration?**

At the time of initial submission, we listed 12 plumes. However, similar plumes do not seem to need to be listed multiple times. Therefore, we chose the typical six plumes and discussed in detail the process of selecting the peak and background values. The data of these six plumes are only suitable for our discussion method, and have no other particularities. In addition, we supplemented the results of all 23 monitoring experiments in the manuscript.

[Figure]

**Figure 6. Comparison between the true values of FSC (x-axis) against the estimated values of FSC (y-axis) of 23 times measurement.**

As shown in Fig 6, the FSC in our experiments was mainly at a level of 0.035% (m/m) to 0.24% (m/m) (only one measurement of 0.37% (m/m), not enough for reference). The deviation of the estimated FSC value calculated using the proposed method was within 0.03% (m/m), although there was some uncertainty. Considering the uncertainties listed in section 3.3, the proposed method provides accurate results. Overall, the estimated FSC is smaller than the true value in many cases. This is because 1–19% of the sulfur in the fuel is emitted in other forms, possibly $SO_3$ or $SO_4$.

**On the other hands, the main strength of the work that the authors compared their plume measurements with the chemical analysis of the fuel. The authors did not mention the biggest challenge of the technique, namely how can we synchronize the time variations of two different measurements ($SO_2$ and $CO_2$) in order to calculate their ratio. In case of broader plume parcels (and thus longer measurement time) during conventional sniffing technique the uncertainty of the integrals are negligible, so more or less exact ratio can be calculated. On the other hands, the sensing technique provides narrow peaks of both components where the uncertainty of the integral is significant. How can the authors describe the differences in the time variation of the two components (see the figures on Pg10)?**

Yes, this is indeed a key technical problem we encountered. We have added the following explanation:

The response time of both sensors is less than 1s. Even if the sampling rates of the two sensors are set to be consistent, the two sensors cannot be completely synchronized. This makes it difficult to calculate the ratio of $SO_2$ and $CO_2$. Our approach is that the sensor sends the average measurement value of the last 10 s to the receiver at an interval of 10 s. Therefore, the interval of integration in Eq. (1) is 10 s. We determined that taking the mean of measurements directly or at shorter intervals leads to too many narrow peaks in one measurement process. This makes it difficult to select the peak value, and the calculation results are unstable. At the same time, the interval should not be set too long, which will make the crest very inconspicuous or too flat. Therefore, we selected 10 s as the empirical parameter value after several experiments.
Also, in the description of result:
After the measurement of plume 5, the communication module was fault when we wanted to adjust sampling rate. We consequently replaced the communication protocol "HTTP protocol" with the "TCP/IP protocol". The main changes involved adjusting the data sampling rate from 10 to 2 s to make it easier to find the peak value (the sensor sends the average measurement value of the last 10 s to the receiver at an interval of 2 s), and the sensors were consequently recalibrated by standard mixture gas.

**In Plume 5 and Plume 6 the $CO_2$ sensor was saturated at different concentrations (1900 vs 5000 ppm). What was the reason?**
The $CO_2$ sensor has a range of 5000ppm. There appears saturation in plume 6 but not in plume 5. It looks like saturated, but it's not saturated. We checked the data as fellow:

| time | $SO_2$ | $CO_2$ |
|------|--------|--------|
| 11:18:14 | 774 | 1122.8 |
| 11:18:25 | 993 | 1616 |
| 11:18:35 | 938 | 1895.2 |
| 11:18:46 | 938 | 1895.2 |
| 11:18:57 | 1049 | 1896.8 |
| 11:19:08 | 1406 | 1893.6 |
| 11:19:18 | 1348 | 1894.8 |
| 11:19:29 | 1078 | 1896 |
| 11:19:40 | 765 | 1886 |
| 11:19:51 | 635 | 1516 |
| 11:20:01 | 603 | 1066.8 |
| 11:20:12 | 576 | 811.6 |

This kind of situation is rare. It is difficult to draw conclusions at this time. We guess that this may be due to sensor uncertainty. In any case, the data in this period were not used as peak values of the plumes as present in the manuscripts.

In the end, we thank the Referee #1 for his/her positive and constructive comments.

---

## Author Comment (AC2)

Answer to Referee #2

We would like to thank Referee #2 for his/her positive and constructive comments and suggestions. We have studied comments carefully and made corrections, which we hope meet with approval. Comments and responses are listed as follows. In order to facilitate the reference to the questions and proposed changes, we use the following color coding:

Color coding:

**Referee comment**

Our answer

Proposed change in manuscript

**Title: Without trying to be negative I would suggest leaving out the words "High-precision".**

Our initial use of the term was based on the fact that UAV measurements can be made more closely to the funnel of ship, to obtain high-precision results. But on reflection, we think the precision will improve further, with the progress of technology and method. "High-precision" is not appropriate; we changed the title as:

"Monitoring of compliance with fuel sulfur content regulations through UAV measurements of ship emissions"

**Abstract: As in the title I would suggest leaving out " high precision" in the last sentence. I would also mention the range of sulfur contents that were encountered in the study i.e. how many non-conformities were encountered. And I would like to mention more explicitly that the deviation of the estimated value for +FSC is less than 0.03% (m/m) at a level of 0.04 % to 0.24 % FSC. Note that in ECA areas, with a limit of 0.1%, an uncertainty of 0.03% is not very good. I would also suggest mentioning that in all cases the estimated FSC was always lower than the actual FSC derived from samples taken on board. This is an important aspect with a strong impact on the usefulness of the method in SECA areas with a 0.1% limit value.**

"High-precision" has been leaving in whole manuscript. The range of sulfur contents is very important for this research, which should be mentioned in the abstract, result and conclusion. These parts have been rewritten. In addition, the discussion about underestimate of FSC has also add in the conclusion.

In abstract:

After more than 20 comparative experiments, the results show that, in general, the deviation of the estimated value for FSC is less than 0.03% (m/m) at an FSC level ranging from 0.035% (m/m) to 0.24% (m/m). Hence, UAV measurements can be used for monitoring of ECAs for compliance with FSC regulations.

In result:

As shown in Fig 6, the FSC in our experiments was mainly at a level of 0.035% (m/m) to 0.24% (m/m) (only one measurement of 0.37% (m/m), not enough for reference). The deviation of the estimated FSC value calculated using the proposed method was within 300 ppm (0.03% (m/m)), although there was some uncertainty.

In conclusion:

In general, the deviation of the estimated FSC value was within 0.03% (m/m) at an FSC level of 0.035% (m/m) to 0.24% (m/m). Because not all the sulfur in the fuel is emitted as $SO_2$, the estimated FSC is smaller than true value in many cases. Therefore, if the maritime department wants to take the estimated value as the basis for the preliminary judgment regarding whether the ship exceeds the emission standard, it needs to set an appropriate threshold and a confidence interval.

**How many non-conformities were encountered.**

In the result, we discuss that:

In addition, when the FSC of the target ship is low, for example, when the fuel used is light diesel fuel, the $SO_2$ observation values were mostly 0. When this happened, according to our experience, the FSC was generally lower than 200 ppm, and the ship was likely to meet the emission requirements.

In the conclusion, we discuss that:

1. In about 10% of the cases, the UAV did not measure the effective background value and peak value. This is mainly caused by the UAV missing the plume during its flight. Therefore, effective methods for finding and navigating to plumes using real-time sensor feeds need to be explored.

2. In about 10% of the cases, the absolute error was more than 0.03% (m/m), and even more than 0.05% (m/m) in rare cases. Unstable concentrations of $SO_2$ or $CO_2$ in the atmosphere just before the measurement may cause such errors. Furthermore, uncertainties, such as sensor uncertainty, exhaust uncertainty, measurement uncertainty, and calculation uncertainty, may hinder accurate measurement.

**Paper: This could be a very useful paper with lots of detail. Especially the level of detail is useful since this is an area with a lot of development and sharing of these new results could very helpful to other scientists. I provide some comments that could help to make the paper a bit clearer in some areas. See my specific comments below.**

Thank you for the comments, we are very encouraged.

**Figure 1: I am not familiar with UAVs and in a first glance I thought the black**

**box mentioned in the text was the large flight case black box below the drone. Page 3 line 16. Not everybody may be familiar with the word "Pod".**

Yes, this "Pod" was designed and customized by us. It's not a commercial product. At first glance it may indeed seem puzzling. We have explained it in more detail in the title of figure 1 and text.

[Figure]

**Figure 1. Image of the modified UAV platform. The black box installed under the UAV is a pod which was designed and customized by us. It carries a gas pump (to collect the ship's exhaust), gas circuit, a filter (to remove water vapor), sensors for $SO_2$ and $CO_2$, a small motor (to provide energy for pumping), a camera, and communication modules.**

In the experiment, we used the MATRICE 600 UAV (SZ DJI Technology Co., Ltd.), and modified it. We designed and customized a special pod, which was installed underneath the UAV, to carry sensors, communication circuit boards, gas circuit systems, and other modules, as shown in Fig.1.

**Page 4 last sentence: electrochemistry method. Electrochemical method?**

Electrochemical method, the term has been rewritten.

**Page 5 line 12-13. These sentences are rather unclear. What is meant with 180 working hours apart? Each 180 working hours? It is not entirely clear what the actual accuracy is if it is 1% full scale.**

This sentence has been rewritten. The accuracy is written as $\pm 0.25$ ppm for $SO_2$ and $\pm 50$ ppm for $CO_2$, respectively.

Sensor calibration is required when the equipment is used daily. The time interval for sensor calibration is three months or when the accumulated working time of the sensor exceeds 180 h. If either of these conditions is met, calibration will be carried

out.

**Page 7 line 16: correction should be corrected. Gradually establishing a quality management system.... Is rather vague what is meant. Please rephrase.**

These sentences have been rewritten.

As for sensor uncertainty, the linear error is negligible and the nonlinearity of the two sensors should be no more than ±1%. It can be corrected through frequent calibrations with standard gases and gradually establishing a quality management system comprising sensor linearity, sensitivity, repeatability, hysteresis, resolution, stability, drift, and other attributes of the minimum requirements.

**Page 7 line 22. Here 200 ppm is mentioned where in other places in the text 0.03 % (300 ppm) is mentioned. This should be explained or there should rather be only one number. Same place: the deviations mentioned in Balzani et al. (2014) were determined at FSC of 1%. It is not clear whether these deviations are still the same at 0.1% FSC. They could be lower at 0.1% FSC content. The authors should mention that or provide more information (which would be useful)**

Yes, the measurement range of the FSC is very important information when discussing the measurement results. We supplement the information of measuring range when we discuss the relative precision. We have made the following description for the "200ppm".

Exhaust uncertainty arises because not all the sulfur in the fuel is emitted as $SO_2$. Preliminary studies showed that 1-19% of the sulfur in the fuel is emitted in other forms, possibly $SO_3$ or $SO_4$ (Schlager et al., 2006, Balzani Lööv et al., 2014). Hence, the assumption that all sulfur is emitted as $SO_2$ yields an underestimation of the true sulfur content in the fuel. Accordingly, this factor needs to be considered when setting the alarm threshold of the FSC. In our experiments, this uncertainty factor led to low FSC estimation results, and the deviation was generally not more than 200 ppm. This prediction is based on the fact that several measurements of some plumes were taken at particular times. Similar calculation results for FSC were obtained, but they were all less than the real value of 100–200 ppm. This is probably because not all the sulfur in the fuel is emitted as $SO_2$. This tendency of underestimation has also been found in previous studies (Johan, R et al. 2017).

[1] Balzani Lööv, J. M., Alfoldy, B., Gast, L. F. L., Hjorth, J., Lagler, F., Mellqvist, J., Beecken, J., Berg, N., Duyzer, J., Westrate, H., Swart, D. P. J., Berkhout, A. J. C., Jalkanen, J.-P., Prata, A. J., vander Hoff, G. R., and Borowiak, A.: Field test of available methods to measure remotely SOx and NOx emissions from ships, Atmos. Meas. Tech., 7, 2597–2613, doi:10.5194/amt-7-2597-2014, 2014.
[2] Johan, R., Conde, V., Beecken, Jörg and Ekholm, J.: Certification of an aircraft

and airborne surveillance of fuel sulfur content in ships at the SECA border, CompMon (https://compmon.eu/), 2017.

[3] Schlager, H., Baumann, R., Lichtenstern, M., Petzold, A., Arnold, F., Speidel, M., Gurk, C., and Fischer, H.: Aircraft-based Trace Gas Measurements in a Primary European Ship Corridor, proceedings TAC-Conference, 83–88, 2006.

**Page 7 last paragraph. To me it is not clear how errors in determination of the peak height is propagated in the total error and it is not clear how this is done. The error of 300 ppm is (it seems) related to the comparison with the on-board samples. And not from error propagation analysis as far as I can tell. It would be nice to show the error propagation numbers as well and see how well these two approaches match. In general, I think that the uncertainty discussion could be more quantitative.**

The results of the FSC are derived from the calculation of four data. Therefore, errors or incorrect selection of these four values can affect the results of the FSC. Therefore, the law of error propagation can explain the uncertainty. I have supplemented the error propagation formula of the FSC formula to illustrate this problem. Currently, the data we can obtain are FSC estimates (derived from four measurements) and FSC true values (derived from chemical validation of the fuel). Currently, only multiple measurements of the same plume or multiple peaks using the same measurement can be used to analyze its uncertainty.

Calculation uncertainty lies in selecting the background and peak values of $SO_2$ and $CO_2$. According to the law of error propagation (widely used in surveying, mapping, and statistics), the relationship between the deviation in the measurement values and that in the FSC can be obtained. The FSC calculation results are functions of independent observations $SO_{2,peak}$, $SO_{2,bkg}$, $CO_{2,peak}$, and $CO_{2,bkg}$ as in formula 1. The relationship between the observation error ($\Delta SO_{2,peak}$, $\Delta SO_{2,bkg}$, $\Delta CO_{2,peak}$, and $\Delta CO_{2,bkg}$) and function error ($\Delta FSC$) can be approximated using the full differential of the function as follows:

$$\Delta FSC = \frac{\partial f}{\partial SO_{2,peak}} \Delta SO_{2,peak} + \frac{\partial f}{\partial SO_{2,bkg}} \Delta SO_{2,bkg} + \frac{\partial f}{\partial CO_{2,peak}} \Delta CO_{2,peak} + \frac{\partial f}{\partial CO_{2,bkg}} \Delta CO_{2,bkg} \quad (2)$$

In our study, this deviation was generally in the order of hundreds of ppm, as explained in section 4.

**Page 9 line 16: "this makes the FSC value relatively larger than that of $CO_2$". It is not clear what is meant here.**

These sentences have been rewritten.

$$FSC[\%] = \frac{S[kg]}{fuel[kg]} = \frac{SO_2[ppm] \cdot A(S)}{CO_2[ppm] \cdot A(C)} \cdot 87[\%] = 0.232 \frac{\int (SO_{2,peak} - SO_{2,bkg}) dt [ppb]}{\int (CO_{2,peak} - CO_{2,bkg}) dt [ppm]} [\%] \quad (1)$$

As in Eq. (1), a higher $SO_2$ peak leads to a higher FSC estimate, while a higher $CO_2$ peak leads to a lower FSC estimate. As discussed in section 3.3, not all the sulfur in the fuel is emitted as $SO_2$, which will result in a lower estimate value. This selection allows the estimate to be relatively close to the true value.

**Page 9 line 6: were synchronized is rather vague. Please explain Page 9 In general, the data treatment is unclear to me. Why are peak values taken to compare $SO_2$ and $CO_2$? Or is it the surface area? The S-content may be derived from any set of concentrations. Taking the pea area ins just of way of averaging. It seems to me now that the peak position and its height is depending on the performances of the sensors (especially response time) and the accidental position in the plume. This could lead to uncertainties especially if the peak height only is used. This should be explained better. Especially the "approach" could be elaborated more. Sometimes I am in doubt whether peak means the highest point in the concentration or the peak area.**

We made the following explanation in the manuscript:

The response time of both sensors is less than 1s. Even if the sampling rates of the two sensors are set to be consistent, the two sensors cannot be completely synchronized. This makes it difficult to calculate the ratio of $SO_2$ and $CO_2$. Our approach is that the sensor sends the average measurement value of the last 10 s to the receiver at an interval of 10 s. Therefore, the interval of integration in formula (1) is 10 s. We determined that taking the mean of measurements directly or at shorter intervals leads to too many narrow peaks in one measurement process. This makes it difficult to select the peak value, and the calculation results are unstable. At the same time, the interval should not be set too long, which will make the crest very inconspicuous or too flat. Therefore, we selected 10 s as the empirical parameter value after several experiments.

Also, in the description of result:
After the measurement of plume 5, the communication module was fault when we wanted to adjust sampling rate. We consequently replaced the communication protocol "HTTP protocol" with the "TCP/IP protocol". The main changes involved adjusting the data sampling rate from 10 to 2 s to make it easier to find the peak value (the sensor sends the average measurement value of the last 10 s to the receiver at an interval of 2 s), and the sensors were consequently recalibrated by standard mixture gas.

**Page 9 in general: what exactly is "selected". This should be made clear. Now it seems a bit arbitrary. Of course, full range values are not used. But what are dramatic changes? Would be useful to explain.**

These sentences have been rewritten:

The global maximum values are selected as peak values for calculating the FSC.

The peak values resulting from dramatic changes (for instance, the change in $CO_2$ exceeded 500 ppm, or $SO_2$ changes by more than 500 ppb) in continuous observations are ruled out. This may be because of exhaust uncertainty.

**Page 9 line 21: 300 ppm at what level??**

These sentences have been rewritten:

However, the final deviation generally does not exceed 0.03% (m/m) at an FSC level of 0.04% (m/m) to 0.24% (m/m).

**Page 10: Figure 5. Sometimes background values of $SO_2$ are 400 ppb? That is very high. Why not subtract the background? Also in plume 6 the background seems to fluctuate very much. makes interpretation of peaks uncertain. please discuss.**

We have made the following discussion in the manuscript:

The background value of $CO_2$ in plumes 1-4 exceeded 300 ppm, but the global background $CO_2$ was approximately 400 ppm. Meanwhile, the background value of $SO_2$ exceeded 400 ppb at some time. This was due to sensor calibration, which did not affect the final result. This kind of situation did not happen again after we recalibrated the sensors by standard mixture gas. In some cases, background values seem to fluctuate very much. This is mainly because the UAV took off from the dock, where multiple ships were berthed and wind speeds were high. Therefore, we used the flight procedure given in section 3.1 to minimize this impact.

**Page 11 table. Why is not a graph provided? Such as true value (x-axis) against estimated value (y-axis). Then also a correlation coefficient could be calculated. Also a good measure of quality. In general: The results section could improve in clarity if some structure was used: data treatment; FSC observed etc. For example, the issues with sampling rate etc. (page 1 top) are perhaps important but mixed here with the results. To increase clarity this could be treated separately Conclusions High precision is not reasonable to state in view of the rather large underestimations.**

This suggestion is very helpful; we added the result graph of all the data.

[Figure]

**Figure 6. Comparison between the true values of FSC (x-axis) against the estimated values of FSC (y-axis) of 23 times measurement.**

As shown in Fig 6, the FSC in our experiments was mainly at a level of 0.04% (m/m) to 0.24% (m/m) (only one measurement of 0.37% (m/m), not enough for reference). The deviation of the estimated FSC value calculated using the proposed method was within 0.03% (m/m), although there was some uncertainty. Considering the uncertainties listed in section 3.3, the proposed method provides accurate results. Overall, the estimated FSC is smaller than the true value in many cases. This is because 1–19% of the sulfur in the fuel is emitted in other forms, possibly $SO_3$ or $SO_4$.

**In general: The results section could improve in clarity if some structure was used: data treatment; FSC observed etc. For example, the issues with sampling rate etc. (page 1 top) are perhaps important but mixed here with the results. To increase clarity this could be treated separately**

The structure has been adjusted. 4.Results: 4.1 Data treatment; 4.2 FSC estimation.

**Page 12: in Conclusions something might be said on the effect of $SO_3$ and $SO_4$ Specific comments:**

As mentioned above, we explained it in the results. At the same time, we make the following explanation in the conclusion.

The estimated results were compared with the FSC values determined at certified laboratories. In general, the deviation of the estimated FSC value was within 0.03%

(m/m) at an FSC level of 0.035% (m/m) to 0.24% (m/m). Because not all the sulfur in the fuel is emitted as $SO_2$, the estimated FSC is smaller than true value in many cases. Therefore, if the maritime department wants to take the estimated value as the basis for the preliminary judgment regarding whether the ship exceeds the emission standard, it needs to set an appropriate threshold and a confidence interval.

**I am not a native speaker, but the English seems fine with me in general. Some specific text could be altered: - on ships the "chimney" is often called the "funnel" - "ship" is normally "vessel". - Culled is not a word that is often used Page 8 line 3: English: none of the monitored ships were fitted with exhaust cleaning equipment**

The overall language of the manuscript has been enhanced; thus, any language and grammar mistakes have been corrected to the greatest extent possible.

Some words were changed as follows:

"Chimney" or "funnel": funnel.

"ship" or "vessel": In the relevant literatures, "ship" seems to be used more frequently.

"Culled": Replace with "be ruled out"

"none of the monitored ships were fitted with exhaust cleaning equipment": It has been changed.

In the end, we thank the Referee #2 for his/her positive and constructive comments.

---

## Author Response (AR2)

Answer to Referee #1

We would like to thank Referee #1 for his/her positive and constructive comments and suggestions. We have studied comments carefully and made corrections, which we hope meet with approval. Comments and responses are listed as follows. In order to facilitate the reference to the questions and proposed changes, we use the following color coding:

Color coding:

**Referee comment**

Our answer

Proposed change in manuscript

**The authors did lots of efforts to correct the manuscript and most of the reviews' comments have been addressed. However, I have a serious concern about the calculation of the fuel sulfur content. They keep calculating the FSC from peak values instead of calculating the ratios of the areas under the S and $CO_2$ time plot. This input a huge uncertainty in the method. I am, as an experimentalist, not fully convinced that I could believe in those ratios even if the calculated FSC is close to the laboratory given value.**

**For this reason, I can't recommend the paper for publication until the authors can prove that the peak value calculation is correct, for example, all the peak values in one measurement give the same FSC with an acceptable statistic.**

Yes, I agree with this viewpoint. Our approach is exactly to use "the ratios of the areas" rather than "peak values". As is written in the manuscripts:

$$FSC[\%] = \frac{S[kg]}{fuel[kg]} = \frac{SO_2[ppm] \cdot A(S)}{CO_2[ppm] \cdot A(C)} \cdot 87[\%] = 0.232 \frac{\int(SO_{2,peak} - SO_{2,bkg})dt[ppb]}{\int(CO_{2,peak} - CO_{2,bkg})dt[ppm]}[\%] \qquad (1)$$

The response time of both sensors is less than 1s. Even if the sampling rates of the two sensors are set to be consistent, the two sensors cannot be completely synchronized. This makes it difficult to calculate the ratio of $SO_2$ and $CO_2$. Our approach is that the sensor sends the average measurement value of the last 10 s to the receiver at an interval of 10 s. Therefore, the interval of integration in Eq. (1) is 10 s. $\int(SO_{2,peak} - SO_{2,bkg})dt\,[ppb]$ are just the area under the S time plot, and the interval of integration is 10 s. Then, formula (1) can be rewritten as

$$FSC[\%] = 0.232 \frac{\int(SO_{2,peak} - SO_{2,bkg})dt[ppb] \Big/ 10}{\int(CO_{2,peak} - CO_{2,bkg})dt[ppm] \Big/ 10}[\%] \approx 0.232 \frac{AVG(SO_{2,peak}) - AVG(SO_{2,bkg})}{AVG(CO_{2,peak}) - AVG(CO_{2,bkg})}[\%] \qquad (2)$$

AVG ( ) is the calculate function for average measurement value. Therefore, the data in the manuscript are the average value of measurement in 10s. Through formula (2), it is just "the ratios of the areas under the S and $CO_2$ time plot".

In fact, we have also proved this viewpoint in the experiment. We increased the sampling rate to 1 s and send the original value of measurement to the receiver in the second-generation pod (as shown in Figure 1) in April, 2019.

[Figure]

Figure 1 The second-generation pod (the volume gets smaller)

[Figure]

Figure 2 Measurement value of the second-generation pod

As shown in Fig. 2, the average values are obviously more stable than the original values, and it is easier to determine the peak value. If the original value is used, the data within certain time range needs to be selected to reduce uncertainty. But the result came out the same as that calculated with the average value.

In fact, in the calculation method with area ratio, the start time and end time of the area are also need to be confirmed. The problem of uncertainty remains. It is a process similar to that of peak value selection method and confirming the time span of integral described in this manuscript. Therefore, "the ratios of the areas" is necessary and can be calculated by average values of measurement. Nevertheless, we used the flight procedure given in section 3.1 and the selection method of peak values to minimize the impact of uncertainties. We believe the sharing of these results could helpful to other researchers. As the comments of Referee #2, "It should be noted that the accuracy of the results of monitoring is a difficult issue and the accuracy estimates in literature may not always be comparable."

Finally, if necessary, I can provide all the data and flight recorders (stored in a database) to demonstrate the effect of this method in 23 experiments.

In the end, thanks again for the positive comments.

Answer to Referee #2

We would like to thank Referee #2 for his/her positive and constructive comments and suggestions. We have studied comments carefully and made corrections, which we hope meet with approval. Comments and responses are listed as follows. In order to facilitate the reference to the questions and proposed changes, we use the following color coding:

Color coding:

**Referee comment**

Our answer

Proposed change in manuscript

**Page 1 Line 20 Perhaps there are more recent data (references) on the contribution of shipping then those from 2010. Especially since there is mention of "rapid development"**

Two references have been added.

Liu et al. (2016) reported that East Asia accounted for 16% of global shipping CO2 emissions in 2013, which was an increase compared to only 4–7% in 2002–2005. In the research of Russo et al. (2018), who evaluated the contribution of shipping to overall emissions over Europe, this sector was found to represent on average 16%, 11%, and 5% of the total $NO_x$, $SO_x$, and $PM_{10}$ emissions, respectively.

Reference:
Russo, M. A. , LeitãO, J. , Gama, C. , Ferreira, J. , & Monteiro, A. . (2018). Shipping emissions over europe: a state-of-the-art and comparative analysis. Atmospheric Environment, 177, 187-194.
Liu, H., Fu, M., Jin, X., Shang, Y., Shindell, D., Faluvegi, G., Shindell, C., and He, K.: Health and climate impacts of ocean-going vessels in East Asia, Nat. Clim. Change., 6, 1037-1041, 10.1038/nclimate3083, 2016.

**Page 2 line 11 English: the FSC content rather than FSC limit. And perhaps it should not say must not exceed but may not exceed.**

This sentence has been rewritten.

The FSC limit was set to 0.1% (m/m) beginning in 2015.

**Page 2 line 26 English: check the sentence starting with Optical methods. Unclear.**

This sentence has been rewritten and added the reference.

Optical methods analyze the variation of the light properties after interaction with the exhaust plume and allow, if the local wind field is known, to determine the emission rate of $SO_2$. The simultaneous measurement of $CO_2$ and $SO_2$ emissions at a routine basis with these systems is unrealistic at the moment (Balzani Lööv et al., 2014).

Balzani Lööv, J. M., Alfoldy, B., Gast, L. F. L., Hjorth, J., Lagler, F., Mellqvist, J., Beecken, J., Berg, N., Duyzer, J., Westrate, H., Swart, D. P. J., Berkhout, A. J. C., Jalkanen, J.-P., Prata, A. J., vander Hoff, G. R., and Borowiak, A.: Field test of available methods to measure remotely SOx and NOx emissions from ships, Atmos. Meas. Tech., 7, 2597–2613, doi:10.5194/amt-7-2597-2014, 2014.

**Page 2 Line 29 English: for calculating change to: to calculate**
**Page 2 Line 32 English: ship's plume rather than ship plume**
10  **Page 3 English: more accuracy estimation changes to more accurate estimation**

These sentences have been rewritten.

15  **Page 3 lines 10 etc. It should be noted that the accuracy of the results of monitoring is a difficult issue and the accuracy estimates in literature may not always be comparable. Please make a note on that.**

Yes, I have added a note as fellow in the manuscript.

It is important to note that the accuracy of the results of monitoring is a difficult issue to address, and the accuracy of estimates in the literature may not always be comparable. For ideal comparison results, one would need to board the ship to take fuel samples, which is particularly difficult for sailing ships.

**Page 3 the section starting line 15 with Ship emission measurements. It is perhaps not the best option to start here with the distinction between land-based methods and others. I would suggest to start with this subject (i.e. land based), add fuel sampling and then address all issues on accuracy etc. that's to me seems a more logic order.**

This paragraph is the introduction of ship emission measurements according to five different platforms. Land-based and airborne-based methods are the two main platforms at present. As such, the most suitable approach for monitoring compliance is to employ "sniffer" measurements taken by airborne. However, the cost of airborne is high. Therefore, it is useful to research the ship emission measurements
35  by UAV.
It is the writing logic of this paragraph.
I'm not sure I fully understand the suggestion. Because the accuracy has been talked about when introducing optical methods and "sniffing" methods in the previous paragraph (introduction of ship emission measurements according to optical methods and "sniffer" methods).
40  However, I think the logic of the section may not be very clear, and I have modified this paragraph to make it more logical.

[revised manuscript text omitted]

**Page 3 line 21 English: collect exhaust gas. This change is also needed in other parts of the Text.**
**Page 3 line 26 English: taken from aircraft and not airborne.**

These sentences have been rewritten.

**Figure 1 legend: filter to remove water vapor? or is that a dryer (nafion)?**
25 It is a gadget to remove water vapor, as follows:

[Figure]

Figure 1 Different types of gadget

**Figure 2 is a nice picture, but it is not very illustrative. There is more than one exhaust and there is no visible plume. Please consider using another picture.**
30 I have checked my phone and some pictures are as Fig. 2. Because the laboratory is located inside the Shanghai Maritime University, exhaust gas is cleaned before discharge in most cases. It is rare to see a plume. The picture used in the manuscript seems to be the best choice.

[Figure]

[Figure]

35 Figure 2 Pictures of the automatic engine room laboratory of Shanghai Maritime University.

**Page 5 If the Matrice instrument is modified, perhaps something should be said on what has been done. Or say: small modifications.**

Modifications mainly includes add the bracket and power interface for the pod, "small modifications" are appropriate, it has been modified.

In the experiment, we used the MATRICE 600 UAV (SZ DJI Technology Co., Ltd.) with a few small modifications.

**Page 6 I thought the $SO_2$ sensor was rather new. Yet reference is made to Hodgson 1999. Is that correct? Please check.**

According to the original suggestion requirements that "give more specific information on how the electrochemical cell mechanism works for $SO_2$". The reference herein is used to illustrate the characteristics of electrochemical sensors of $SO_2$. It is not for this particular sensor that we used.
I have modified the sentence as:

The $SO_2$ electrochemical sensor has the advantages of low power consumption, small size, light weight, and high precision. In addition, **this type of sensor** is capable of measuring $SO_2$ at a low ppb range (Hodgson et al., 1999).

Hodgson, A. W. E., Jacquinot, P., and Hauser, P. C.: Electrochemical Sensor for the Detection of $SO_2$ in the Low-ppb Range. Analytical Chemistry, 71(14), 2831–2837, doi:10.1021/ac9812429, 1999.

**Page 7: at what distance from the ship is the background determined? This may vary but what are ranges.**

More than 50 m away from the ship's smoke. As described in the manuscript: "3. The UAV takes off vertically and rises to an altitude of 100 m (the first measurement point) for 3 min to determine the background value of $SO_2$ and $CO_2$.". I have added information as fellow.

3. The UAV takes off vertically and rises to an altitude of 100 m (the first measurement point) for 3 min to determine the background value of $SO_2$ and $CO_2$. The take-off position is usually on the dock and is more than 50 m away from the ship's smoke.

**Page 8 line 2: English: flied change to flew**

It has been modified.

**Pag 2 line 3. Why not determine the background after the peak as well? And see how it changes**

**the results (or not). Please discuss.**

It seems to be at Pag 8 line 3.
It's been discussed in the manuscript as fellow:
"Observed $SO_2$ and $CO_2$ values returned to background levels, but they were not used as background values. Residual gas in the airway needed to be discharged by the gas pump before the next collection".

**Page 8 discussion on sampling interval. It does not seem to be completely solved by setting all intervals to 10 second. This is an outcome of the experiments.**

According to the first round of comments. I added this discussion on sampling interval to explain "how can we synchronize the time variations of two different measurements ($SO_2$ and $CO_2$) in order to calculate their ratio."

$$FSC[\%] = \frac{S[kg]}{fuel[kg]} = \frac{SO_2[ppm] \cdot A(S)}{CO_2[ppm] \cdot A(C)} \cdot 87[\%] = 0.232 \frac{\int (SO_{2,peak} - SO_{2,bkg})dt[ppb]}{\int (CO_{2,peak} - CO_{2,bkg})dt[ppm]}[\%] \qquad (1)$$

The interval of integration is 10 s in Eq. (1), which is a parameter.

As discussed in section of experimental results. "After the measurement of plume 5, we adjusted the data sampling rate from 10 to 2 s to make it easier to find the peak value (the sensor sends the average measurement value of the last 10 s to the receiver at an interval of 2 s)." But the interval of integration in Eq. (1) is still 10 s.
I agreed to the fact that "It does not seem to be completely solved by setting all intervals to 10 second". In fact, we increased the sampling rate to 1 s and send the original value in the second-generation Pod in April, 2019. After a few experiments of changing the interval of integration, the accuracy did not seem to improve significantly. The interval of 10 s is appropriate to achieve accurate results relatively.

**Page 8 discussion on what is called "exhaust uncertainty" is difficult. In my opinion this is a completely different uncertainty and should be discussed on a different location in the paper.
-We measure the composition of the exhaust gas and derive from that the S content in the fuel. There are errors: caused by inadequate sampling, calibration, errors subtraction of the background etc. An attempt should be made to estimate the total error in the measured fuel content.
That is one side of the comparison:
-on the other side there is fuel sampling with its errors (perhaps), and the representativity of the sample. Is the fuel sample representative for the exact moment of the measurement with the UAV? Is there a chance that the ship may have switched fuel.? These are additional uncertainties (that play a role in European harbors: switching of fuels. But no need to go into that. Ok we assume that the sample is representative is the right sample.
Then the last question is:
-is all Sulphur converted to $SO_2$ or also to other species. In Balzani et al a number of 6 % Sulphur s mentioned that is not converted. A correct sniffer method would then automatically underestimate the fuel content by 6%. This was at a S content of 1%. It is unclear whether this percentage is still to be expected at 0.5 % Sulphur. Please make a note of that. The 6% is rater**

**uncertain and probably depends on several parameters.**

**Therefore, I think that the exhaust uncertainty should not be discussed in the series of errors because it is not an error. It may cause unexpected differences, but it is not an error. A flawless monitoring method would discover this difference. Discuss this issue in the final comparison**

Yes, I agree with the viewpoint that exhaust uncertainty is different. I have put the introduction of "exhaust uncertainty" at the end of section 3.3 (uncertainties), and the discussion of "exhaust uncertainty" in our experiment at section 4.2 (FSC estimation).
In section 3.3:

Exhaust uncertainty arises because not all the sulfur in the fuel is emitted as $SO_2$, which is a completely different uncertainty. Preliminary studies showed that 1-19% of the sulfur in the fuel is emitted in other forms, possibly $SO_3$ or $SO_4$ (Schlager et al., 2006, Balzani Lööv et al., 2014). Hence, the assumption that all sulfur is emitted as $SO_2$ yields an underestimation of the true sulfur content in the fuel. Accordingly, this factor needs to be considered when setting the alarm threshold of the FSC.

In section 4.2:

As shown in Fig 6, the FSC in our experiments was mainly at a level of 0.035% (m/m) to 0.24% (m/m) (only one measurement of 0.37% (m/m), not enough for reference). Overall, the estimated FSC is smaller than the true value in many cases. This could be due to the exhaust uncertainty that not all the sulfur in the fuel is emitted as $SO_2$. In our experiments, this uncertainty factor led to low FSC estimation results, and the deviation was generally not more than 200 ppm. This prediction is based on the fact that several measurements of some plumes were taken at particular times. Similar calculation results for FSC were obtained, but they were all less than the real value of 100–200 ppm. This tendency of underestimation has also been found in previous studies (Johan, R et al. 2017).

"switching of fuels" is an important factor. In our experiment, one person operated the UAV to monitor the plume. Two maritime law enforcement officers board the ship to collect fuel samples at the same time. Both processes took about 10-20 minutes. As a result, the sample can be thought as the right sample.

**Pag 9. Put the remark on the absence of ships with exhaust cleaning in the sample somewhere else. Perhaps in the conclusion.**

This statement is also intended to remove uncertainty of this factor. I put it in the uncertainty analysis.

**Page 10 line 4 English: ship smoke: change to the ships plume**

It has been modified.

**Page 10 selection method: number 3……No problem ruling out those cases, but I wonder how exhaust uncertainty could explain this. As was mentioned earlier in the text: this is because only**

**94 % of the Sulphur is oxidized to SO₂.**

"The peak values resulting from dramatic changes" means that the percentage of $SO_2$ and $CO_2$ in the plume is not stable at the time of measurement. "Exhaust uncertainty arises because not all the sulfur in the fuel is emitted as $SO_2$. Preliminary studies showed that 1-19% of the sulfur in the fuel is emitted in other forms, possibly $SO_3$ or $SO_4$ (Schlager et al., 2006, Balzani Lööv et al., 2014)".
Therefore, I guess the dramatic changes in the percentage of $SO_2$ and $CO_2$ is probably due to changes in the conversion rates (S to $SO_2$ or C to $CO_2$).
But that's just a guess of me. So, I used the word "may".
This **may** be because of exhaust uncertainty.
In addition, sensor uncertainty and unstable concentrations of $SO_2$ or $CO_2$ in the atmosphere may also cause such case. I have added.

This may be because of sensor uncertainty, exhaust uncertainty or unstable concentrations of $SO_2$ or $CO_2$ in the atmosphere.

Schlager, H., Baumann, R., Lichtenstern, M., Petzold, A., Arnold, F., Speidel, M., Gurk, C., and Fischer, H.: Aircraft-based Trace Gas Measurements in a Primary European Ship Corridor, proceedings TAC-Conference, 83–88, 2006.
Balzani Lööv, J. M., Alfoldy, B., Gast, L. F. L., Hjorth, J., Lagler, F., Mellqvist, J., Beecken, J., Berg, N., Duyzer, J., Westrate, H., Swart, D. P. J., Berkhout, A. J. C., Jalkanen, J.-P., Prata, A. J., vander Hoff, G. R., and Borowiak, A.: Field test of available methods to measure remotely SOx and NOx emissions from ships, Atmos. Meas. Tech., 7, 2597–2613, doi:10.5194/amt-7-2597-2014, 2014.

**Page 11 first sentence This will make the FSC relatively larger that of CO₂. Unclear what is meant.**

[Figure]

Fig. 3 screenshots

As shown in Fig. 3:
There is a small deviation between the time point of the peak values for $SO_2$ and $CO_2$. Therefore, there is two ways to select the time point of peak value: (1) the time point of $SO_{2,peak}$. (2) the time point of $CO_{2,peak}$. Different selection methods result in small differences in FSC estimates.

$$FSC[\%] = \frac{S[kg]}{fuel[kg]} = \frac{SO_2[ppm] \cdot A(S)}{CO_2[ppm] \cdot A(C)} \cdot 87[\%] = 0.232 \frac{\int (SO_{2,peak} - SO_{2,bkg})dt[ppb]}{\int (CO_{2,peak} - CO_{2,bkg})dt[ppm]} [\%] \quad (1)$$

As in Eq. (1), The S in the numerator, C (fuel) in the denominator. We select the time point at peak of $SO_2$. "This will make the FSC value relatively larger than that of $CO_2$. As in Eq. (1), a higher $SO_2$ peak leads to a higher FSC estimate, while a higher $CO_2$ peak leads to a lower FSC estimate. As discussed in section 3.3, not all the sulfur in the fuel is emitted as $SO_2$, which will result in a lower estimate value. This selection allows the estimate to be relatively close to the true value."

**Figure 5 This is the core of the paper. I have many comments, but will not all give them:**

**(1)-Plume 1. There is an additional SO₂ source (and not CO₂ obviously) That is strange. Often SO₂ comes from fossil fuel burning. So if the SO₂ reses often CO₂ rzes as well. Where does the SO₂ come from?. Or is it a drift or cross sensitivity in the SO₂ sensor? Please comment. Nevertheless, I think the interpretation is OK.**

**(2)-Plume 2 Same on the SO₂ background. Poor quality expected**

**(3)-Plume 6 seems good quality. But the maximum value is reached, and it is rejected(?)**

(1-2) I made more comments in the manuscript as fellow:

In some cases, background values seemed to fluctuate greatly. This was mainly because the UAV took off from the dock, where multiple ships were berthed, and wind speeds were high. In addition, the drift or cross sensitivity in the sensors also may have caused interference. Therefore, we used the flight procedure given in section 3.1 and the selection method of picking values to minimize this impact.

(3) The peak values of $CO_2$ at the full range of the $SO_2$

**General remarks:**

**-(1) I would consider not using auto scaling of the Y axis to give the reader the chance to compare the different plumes**

**-(2) The different peaks in the concentrations are related to the distance to the funnel. If you go a bit further away from the funnel perhaps the plumes are more mixed, and no individual peaks are observed and interpretation is easier. Please discuss.**

**-(3) Although the level of detail is nice to inform the reader some structure could still help. Perhaps make a distinction between: good quality examples and poor-quality examples and rejected plumes, cases with high Sulphur content, case with low Sulphur content and then don't use autoscaling of the y axis. Also show how well the selection procedure works in the presented cases.**

**-(4) It is stated that the deviation is within 300 ppm is. But it is not entirely clear how the best value is selected. Please discuss the selection of the best value in relation to the error estimates on page 9 (equation 2 and the outcome of that)**

(1) I tried to not use the auto scaling of the Y, but the results didn't seem to be ideal. Take the plume 2 and plume 6 as examples: Y axis of $CO_2$ is 800 ppm and $SO_2$ is 800 ppb in plume 2. In plume 6, the value is 6000 ppm and 4000 ppb, respectively. If we set the Y axis as fixed axes (6000 ppm and 4000 ppb for $CO_2$ and $SO_2$, respectively), then the peak value of plume 2 is difficult to distinguish as show in Fig 4.

[Figure]

Fig 4. Plume 2 in fixed axes

(2) According to our experience, the gas concentration may exceed the range of sensor when the UAV is very close to the funnel. However, it is difficult to establish a clear correspondence between distance and measurement value. Wind speed and direction may cause interference. In addition, different ships have different flue lengths. I guess this problem may be testable in the laboratory.

(3) Good advice, I have made corresponding modifications in section of experiment.

(4) I'm not sure I understand the question exactly.

How to select the peak and background values are listed in Table 2, Fig. 5 and corresponding discussion in the manuscript. When the peak and background values are determined, the best value is confirmed. Whether this means that there are still many options after the peak or background is confirmed? No, it is confirmed. As written in the manuscript:

"According to the flight record, the minimum values of $SO_2$ and $CO_2$ collected at the first measurement point are selected as the background values." "The occurrence time of peak values in $SO_2$ and $CO_2$ are compared, and then the simultaneous peaks and almost simultaneous peaks (no more 20 s) are retained. If there is a small deviation between the time point of the peak values for $SO_2$ and $CO_2$, we select the time point at peak of $SO_2$."

$$\Delta FSC = \frac{\partial f}{\partial SO_{2,peak}}\Delta SO_{2,peak} + \frac{\partial f}{\partial SO_{2,bkg}}\Delta SO_{2,bkg} + \frac{\partial f}{\partial CO_{2,peak}}\Delta CO_{2,peak} + \frac{\partial f}{\partial CO_{2,bkg}}\Delta CO_{2,bkg} \qquad (2)$$

$$\Delta SO_{2,peak} = TrueValue(SO_{2,peak}) - MeasurementValue(SO_{2,peak}) \qquad (3)$$

Although the true value of FSC and the measurement value of $SO_2$ can be obtained. However, the true value of $SO_2$ is unknown. It is often replaced by the average of multiple measurements which can be got in simulations or laboratory experiments. In our experiment, formula 2 only shows the relationship between the deviation in the measurement values and that in the FSC.

**Page 13 line 15. Please rephrase to: This could be due to Sulphur emitted in other forms. This is not proven or so in this paper.**

**-(1) The choice of peaks that are selected in the end remains a bit arbitrary. Picking the highest perhaps leads to the best comparison but why? The authors should defend this choice. Or illustrate what happens when other choices are made**

**-(2) Perhaps in the table the three peaks could be averaged, and a standard deviation in the calculated Sulphur content could be presented. Just to illustrate how large errors may be.**

**-(3) And it would be very nice if the error calculated from equation 2 is given for each sample. This could show why difference between measurements by UAV and fuels samples are sometimes larger than other times**

**-(4) If the background values for $CO_2$ can be used it is perhaps better to carry out a recalibration afterwards. Presenting $CO_2$ concentrations far below 400 is not wanted. Or use an arbitrary scale (?). pleas change.**

I have made corresponding modifications.

(1) "Picking the highest perhaps leads to the best comparison but why?"

My guess is as follows:

The maximum values are likely to have been measured in the center of the ship's plume. At that location, the measurement value is relatively stable, and the probability of interference from other factors is lower.

I think this can be tested by detailed experiments in the laboratory. Nevertheless, the results of other choices are list in Table 2 in the manuscript. It can be seen that this approach can indeed achieve more accurate results according to the experiments. This idea has been proved to some extent.

(2) "how large errors" is listed in table 2. It can be seen that the number of peaks is usually 1-3. According to the selection method of peak values, some peaks need to be ruled out., only one or two peaks can be retained. Standard deviation is hard to calculate.

If there are multiple alternative results in one experiment, then if the deviation between the results is small, then it seems that the results are relatively stable; otherwise, a large deviation may occur.

However, based on the current experimental results (23 samples, only one peak in sometimes), it does not seem that the average value can get more accurate results.

(3) Yes, the biggest obstacle is that the cost of experiment is too high. "the accuracy of the results of monitoring is a difficult issue and the accuracy estimates in literature may not always be comparable."

(4) Yes, after the measurement of plume 5, the sensors were consequently recalibrated by standard mixture gas.

**Page 12 line 10. It is not clear what is meant with interpolation ratio.**

The sentence was modified as follows:

Nonetheless, Eq. (1) was used to calculate the ratio of sulfur dioxide difference to carbon dioxide difference, and it therefore does not affect the final calculation results.

**Pag 14 line 13. Significant contribution to the literature. I would refrain from such statements. That is for the readers to decide. Perhaps state that this research is part of a process towards reliable equipment to monitor FSC in SECA.**

Yes, I have made corresponding modifications.

**Page 15 point 2. Is it possible to sate when errors are large? This could be used to identify poor measurements. And it would be helpful.**

Yes, I have made corresponding modifications.

Poor-quality data or rejected plumes may derive from these situations.

**I think the battery life is important but a rather detail and perhaps not for the conclusions but somewhere in the technical descriptions.**

Yes, I have made corresponding modifications.

**Abstract:**
**- change the order of first two sentences.**
**- describe the principles of the method in a few words/lines**
**- Use 23 samples rather than more than 20**

Yes, I have rewritten the abstract.

Air pollution from ship exhaust gas can be reduced by the establishment of Emission Control Areas (ECAs). Efficient supervision of ship emissions is currently a major concern of maritime authorities. In this study, an Unmanned Aerial Vehicle (UAV)-based measurement system for exhaust gas from ships was designed and developed. Sensors were mounted on the UAV to measure the concentrations of $SO_2$ and $CO_2$ in order to calculate the fuel sulfur content (FSC) of ships. Waigaoqiao port in the Yangtze River Delta, an ECA in China, was selected for monitoring compliance with FSC regulations. Unlike in situ or airborne measurements, the proposed measurement system could be used to determine the smoke plume at about 5 m from the funnel mouth of ships, thus providing a means for estimating the FSC of ships. In order to verify the accuracy of these measurements, fuel samples were collected and sent to the laboratory for chemical examination, and these two types of measurements were compared. After 23 comparative experiments, the results showed that, in general, the deviation of the estimated value for FSC was less than 0.03% (m/m) at an FSC level ranging from 0.035% (m/m) to 0.24% (m/m). Hence, UAV measurements can be used for monitoring of ECAs for compliance with FSC regulations.

**Suggestions**
**Please identify weak points in the current approach and provide suggestions for further research and improvements**

It has been added in the conclusion.

[revised manuscript text omitted]

---

## Author Response (AR3)

**amt-2019-29**

Answer to Associate Editor:

We would like to thank Associate Editor for his/her positive and constructive comments and suggestions. We have studied comments carefully and made corrections, which we hope meet with approval.

5  Comments and responses are listed as follows. In order to facilitate the reference to the questions and proposed changes, we use the following color coding:

Color coding:

**Associate Editor comment**

Our answer

10  Proposed change in manuscript

Dear Folkert Boersma,

Thank you for your evaluation and approval of our works. We are very encouraged. According to requirements, we used the Author's response file to make changes (in order to distinguish the last change, this time of changes are marked with yellow color). Response is as follow.

**Abstract, L12: *The* Waigaoqiao port ... was selected**

It has been modified. P4, L13. (The position in this Author's response file, the same below).

**Abstract, L17: "fuel samples were collected *at the same time* and sent"**

20  It has been modified. P4, L17.

**Intro, L26: fuels --> fuel**

It has been modified. P4, L26-27.

25  **P16, L9: "received" is a strange word, do you mean accepted?**

Yes, "accepted" is a more appropriate word. It has been modified. P5, L9.

**P16, L17: The FSC limit was set to 0.1% (m/m) *in those areas* beginning in 2015.**

It has been modified. P5, L17.

30

**P16, L28: cannot --> are not**

It has been modified. P5, L28.

**P18, L21: a UAV to *simultaneously* measure ...**

35  It has been modified. P7, L21.

**P23, L20: "can be obtained" --> is obtained**

It has been modified. P12, L20.

40  **P23, L31: to calculate the *instantaneous* taio of SO2 and CO2**

It has been modified. P12, L31.

**P24, L1: "We determined" --> We found**

It has been modified. P13, L1.

**P24, L9: 'Regarding sensor uncertainty, the nonlinearity „‚' would be a better way to start the sentence.**

It has been modified as follow. P13, L9-10.

Regarding sensor uncertainty, the nonlinearity of the two sensors should be no more than ±1% and the linear error is negligible.

**P25, L1: "a completely different uncertainty" --> please write that this is a systematic uncertainty.**

It has been modified. P14, L1.

**P26, L10-11: "to the" can be removed**

It has been modified. P15, L10-11.

**P26, L11: "peak values *were* not discussed**

It has been modified. P15, L11.

**P26, L26: no more than 20 s apart?**

Yes, it has been modified. P15, L26.

**P27, L9: "dividing line *between* plumes *with* high-sulfur and ..."**

It has been modified as follow. P16, L8-9.

An FSC of 0.01% (m/m) was used as the dividing line between plumes with high-sulfur and low-sulfur content samples.

**P30, L5: "not enough for reference" ... why? It looks like a reliable result to me, explain why it should be discarded.**

[Figure]

Fig. 6, only one measurement (0.37% (m/m)) at the level of 0.24% (m/m) to 0.37% (m/m).

According to previous studies (Balzani Lööv et al., 2014, Van Roy and Scheldeman, 2016a, 2016b) and the Referees' comments, deviations of FSC are not the same at different FSC levels. There was only one measurement (0.37% (m/m)) at the level of 0.24% (m/m) to 0.37% (m/m). This single measurement is not enough to indicate the deviation at this level, although it looks like a reliable result. Therefore, the study concluded that "The deviation of the estimated FSC value was within 0.03% (m/m) at an FSC level of 0.035% (m/m) to 0.24% (m/m)". These data were associated with a relatively high level of certainty.

This part has been modified as follow. P19, L4-6.

As shown in Fig. 6, the FSC in our experiments was mainly at a level of 0.035% (m/m) to 0.24% (m/m). There was one measurement of 0.37% (m/m), especially. However, it is not enough to illustrate the deviation at the level of 0.24% (m/m) to 0.37% (m/m), because deviations of FSC are not the same at different FSC levels.

**P31, L14: "Poor quality data or rejected plumes" ... derive from what sort of situations? Please clarify this sentence.**
These situations are unstable concentrations of $SO_2$ or $CO_2$ and uncertainties as mentioned previously. In addition, I guess "result from" seems more appropriate.
It has been modified as follow. P20, L15-16.

[revised manuscript text omitted]